# Stability of Myeloid Cell Phenotype and Function Across a Broad Age Range in Humans and Cynomolgus Monkeys, and a Dominant Contribution of Humoral Factors in the Control of Bacterial Infection

**DOI:** 10.3390/biomedicines14010071

**Published:** 2025-12-29

**Authors:** Elena V. Lysakova, Marina Y. Burak, Ilya Larin, Sergey A. Chuvpilo, Viktor S. Laktyushkin, Alexander N. Shumeev, Igor E. Pismennyi, Vladimir Y. Toshchakov, Mikhail Y. Bobrov, Stanislav A. Rybtsov

**Affiliations:** Sirius University of Science and Technology, Olimpiyskiy Ave. b.1, 354340 Sirius, Krasnodar Region, Russia; lysakova.ev@talantiuspeh.ru (E.V.L.);

**Keywords:** immunoaging, inflammaging, myeloid bias, senescence, cynomolgus monkeys, infectious diseases, host defense, phagocytes, innate immunity, aging, senescence

## Abstract

**Background**: Immune aging is a complex process involving various cellular changes, such as a myeloid bias, decreased functional activity of immune cells, accumulation of senescent cells, and alterations in serum levels of bactericidal humoral factors. As believed, these changes contribute to increased susceptibility of older adults to infectious diseases. Myeloid cells are considered the first line of defense against bacterial invasion. However, it remains unclear whether the protective functions of myeloid cells diminish in active older adults and whether potential age-related changes are evolutionarily conserved across primates. **Methods**: In this study, myeloid cell populations from peripheral blood and bone marrow of cynomolgus macaques and human peripheral blood were analyzed across a broad age range for phenotypic and functional characteristics, e.g., *E. coli* phagocytosis, secretion of proinflammatory factors, genetic instability, and signs of cellular aging. **Results**: Despite minor interspecies phenotypic differences in granulocyte populations, both the quantity and functions of myeloid cells were remarkably stable during aging in both species. Myeloid cells maintained genetic stability, and high SA-β-Gal activity was observed, likely reflecting metabolic traits rather than age-related changes. Importantly, a predominant and age-independent role of humoral factors, rather than cellular mechanisms, was identified in the initial control of bacterial infection. **Conclusions**: These findings suggest that innate immune functions remain stable for a long time during aging in both species.

## 1. Introduction

Aging of the immune system is accompanied by a quantitative decrease in lymphoid cells, a decrease in their immune functions, while at the same time an increase in the proportion of myeloid cells is observed in mice [1], humans [2] and primates [3]. This is often referred to as the age-related myeloid bias. This immune aging is associated with changes in the hematopoietic stem cell and progenitor pool, which acquires a bias toward preferential production of myeloid progenitors [4,5,6].

However, it is not yet fully understood how the functions of myeloid cells—key components of innate immunity—change in active, healthy, employed individuals across different age groups. It remains uncertain whether myeloid cell function declines as people age. Furthermore, the evolutionarily conserved mechanisms underlying innate immune cell aging across different primate species are still poorly understood. Investigating these processes presents a vital opportunity to identify fundamental, mechanisms of innate immune aging, which are essential for advancing medical and preclinical research.

One of the fundamental hallmarks of tissue aging, as described in the literature, is the increasing genetic instability and the accumulation of senescent cells with age. This buildup has been observed in various tissues throughout the body. Additionally, the process of senescent cell accumulation and tissue aging occurs unevenly across different tissues and cell types [7].

Recent studies have also demonstrated the accumulation of senescent cells among lymphocytes, highlighting the widespread impact of this phenomenon for immune cells [8]. Moreover, DNA repair-deficiency dramatically accelerates the accumulation of senescent cells with age in mice [9].

One of the significant consequences of genetic instability and the accumulation of senescent cells is the disruption of immune function. Genetic instability can be detected through the presence of phosphorylated histone H2AX at serine 139—known as γH2AX—a marker of double-strand DNA breaks and the activation of repair mechanisms. γH2AX plays a critical role in the early stages of the DNA damage response (DDR), which can lead to cell death or trigger the transition to a senescent state. Elevated levels of γH2AX also indicate the presence of persistent, unrepaired double-strand breaks, a hallmark commonly associated with cellular senescence [10,11,12,13,14].

The second key marker of cellular aging is HMGB1 (High Mobility Group Box 1), a non-histone chromosomal protein involved in maintaining chromatin structure and facilitating DNA bending. HMGB1 functions as a DNA chaperone and is predominantly located in the nucleus, where it plays a role in the DNA damage response. The translocation of HMGB1 from the nucleus to the cytoplasm is a significant event, indicating substantial genetic damage and is a feature of cellular senescence. Furthermore, when actively secreted by senescent cells or passively released from damaged cells, HMGB1 acts as a damage-associated molecular pattern (DAMP). By activating inflammatory pathways, it is recognized as one of the key contributors to ‘inflammaging’ (inflammation-driven aging). These processes of DNA damage, senescence, and HMGB1 release can be triggered also by external stress such as viral infections, toxic agents, or radiation [15,16,17].

In addition to these markers, cellular senescence can be evaluated by the expression of SA-β-Gal (senescence-associated β-Galactosidase). The activity of this lysosomal enzyme tends to increase with cell aging, corresponding to an elevated number of lysosomes and their abnormal activity in senescent cells [18].

The combination of these markers with morphological features—such as chromatin condensation, nuclear shape, and cellular structure—provides a more comprehensive characterization of the terminal senescent cellular state in mammalian tissues [19].

Despite the widespread use of γH2AX, HMGB1, and SA-β-Gal markers in cancer and senescent tissue research, their expression profile in innate immune cells has been less extensively studied. However, investigating these markers could provide crucial information about the rate and impact of immune cell aging.

One of the most important functions of the innate immune system, provided by cells of myeloid origin, is the control of extracellular infections through the recognition of pathogen patterns and their elimination via phagocytosis.

When bacteria invade the body, phagocytes recognize their molecular patterns through various Toll-like receptors (TLRs) present on their surface and within endosomes. TLRs can detect components of bacterial cell walls, lipids, and genetic material such as unmethylated DNA. These receptors form complexes with accessory molecules; for example, CD14 is a crucial component of the CD14/TLR4/MD2 complex that specifically recognizes lipopolysaccharide (LPS) from Gram-negative bacteria. Upon LPS binding, the receptor complex undergoes internalization and activates a series of signaling cascades that promote inflammation and phagocytosis. Additionally, a subset of phagocytes expresses Fc receptors (e.g., low-affinity CD16/FcγRIII), which recognize the Fc fragment of antibodies bound to pathogens, opsonize them, and thereby enhance their clearance [20].

Phagocytosis is considered a critical mechanism for controlling extracellular infections and the first line of defense against bacterial invasion [21]. There are conflicting data in the literature regarding age-related changes in phagocytic activity. This inconsistency arises because the term “phagocytosis” also encompasses the ingestion of various objects by cells, including both infectious agents and non-infectious particles. In some studies, age-related alterations in phagocytosis were evaluated by measuring the efficiency of ingestion of sheep red blood cell [22], apoptotic cells [23], or nanoparticles [24].

Additionally, different experimental conditions and diverse cell populations are employed to study phagocytosis, further contributing to the variability in findings.

Most phagocytosis studies employ protocols that involve preliminary cell adhesion or additional differentiation of monocyte-derived macrophages (MDMs). As a result, a decline in the phagocytosis of Staphylococcus aureus by human MDMs has been observed, along with reductions in MDM migration and chemotaxis [23,25,26]. However, this model does not fully represent the innate capacity of circulating blood myeloid cells to control infections. The risk of sepsis increases significantly with age, primarily due to a decline in neutrophil functions such as phagocytosis, degranulation and migration during the early stages of the infection [27].

Other studies have reported that decrease in phagocytic activity is often limited to specific tissues. For example, in mice, a decrease in phagocytosis was observed in peritoneal macrophages, while the phagocytic capacity of bone marrow monocytes and MDMs remained stable with age [28].

The classical object of study for phagocytosis in investigations of the functional activity of myeloid cells within the context of infectious immunity is the opportunistic model bacterium *Escherichia coli* (*E. coli*, a Gram-negative bacterium) [29,30,31].

*E. coli* is widely used in biotechnology and is a natural part of the gut microbiome. However, certain pathogenic strains, especially in immunocompromised individuals, can cause infections. These include urinary tract infections, pneumonia, and gastrointestinal illnesses such as bloody diarrhea, which can progress to sepsis [32,33,34].

On the phagocyte surface, toll-like receptor 4 (TLR4) and its co-receptor CD14 form a complex that recognizes lipopolysaccharide (LPS), a component of the *E. coli* cell wall. CD14 acts to amplify TLR4 signaling. LPS and other bacterial components promote the migration of myeloid cells to the infection site and stimulate the secretion of proinflammatory cytokines [34].

In this study, the term “phagocytosis” was used in the context of infectious immunity and bacterial cell engulfment in the blood and bone marrow of humans and cynomolgus monkeys. To approximate physiological conditions, we employed a protocol for phagocytic analysis in unmanipulated whole blood. When necessary (for example, for SA-β-Gal staining) blood after lysis of erythrocytes was also used for both species.

One of the primary functions of innate immunity is signaling via the secretion of inflammatory factors in response to pathogen components. It is believed that this secretory response may decline sharply with age. However, the issue of secretory aging in myeloid populations remains controversial. Several studies have demonstrated quantitative changes in myeloid cells that secrete inflammatory factors [35], while other reports indicate low levels of cytokine secretion from aged mice upon stimulation of myeloid cells with bacterial components [36]. Additionally, it has been suggested that the overall cytokine response remains relatively unattenuated with age, although low-level chronic inflammation tends to increase over time [37].

In addition to phagocytosis, humoral factors play a significant role in controlling bacterial infections. For example, complement proteins—produced by a variety of cells, including liver cells, epithelial cells, endothelial cells, and immune cells—are key components of innate immunity [38]. Studies in healthy individuals have shown that the activity of the classical and alternative complement pathways increases with age, whereas activity of the lectin pathway decreases. Additionally, concentrations of certain complement components (such as C5, C8, and C9) tend to increase, while factor D levels decrease [39].

Some researchers attribute the diminished infection control observed in elderly individuals to reduced antibody affinity. This decline is influenced by factors such as impaired antibody maturation, a depleted B-cell receptor repertoire, and age-related hyperglycosylation [40,41,42].

In general, the loss of opsonin activity hampers antibody-dependent phagocytosis and cytotoxicity. Concurrently, attenuated complement activity and altered defensin levels contribute to a dysregulated humoral environment. Collectively, these deficits compromise the overall effectiveness of the immune response and increase susceptibility to infections in the elderly [43].

In this study, we investigate whether individuals of working age, pre-retirement, and early retirement exhibit distinct features of age-related innate immune dysfunction that may impair antimicrobial defense. We hypothesize that the age-related shift in hematopoiesis toward myeloid lineages may serve as an adaptation to compensate for reduced functional activity of phagocytes or a diminished contribution of lymphoid cells to immune responses. To evaluate phagocytic activity, a dual-staining *E. coli* preparation was developed, enabling the measurement of phagocytosis in blood suspensions and allowing for the spatial localization of bacteria relative to phagocytes. This method facilitated the study of ex vivo phagocytosis dynamics across various populations in whole blood and bone marrow samples.

The study also conducted a comparative analysis of the contribution of phagocytosis to direct infection control versus the contribution of humoral factors to the elimination of bacterial infection in the bloodstream.

Cynomolgus monkeys are one of the most widely used models in preclinical research, making this study relevant for a broad range of biomedical scientists. A comparison of myeloid populations in blood samples from humans and cynomolgus monkeys, including a previously unstudied group of elderly macaques over 20 years of age, revealed both interspecies similarities and differences in cellular aging processes among primates.

## 2. Materials and Methods

### 2.1. Conjugation of Bacteria with Live/Dead Fixable Violet and Biotin

All procedures were performed under sterile conditions. One day prior to the experiment, 50 μL of the *E. coli* stock suspension was inoculated into 10 mL of LB medium and grown overnight until reaching an optical density 0.6 at 600 nm (in average 1.29 × 10^10^ bacteria per mL). The next day, 500 μL of *E. coli* culture was centrifuged for 2 min at 2350 *g* and resuspended in 250 μL of phosphate-buffered saline (PBS). Subsequently, 250 μL of fixative (10% neutral buffered formalin, Element Company, Saint-Petersburg, Russia) was added and thoroughly mixed by pipetting and vortexing. The samples were incubated for 30 min at room temperature using rotary mixer at 10 rpm. The bacteria were then washed 3 times with 500 μL of PBS via centrifugation for 2 min at 2350 *g* each time. After the final wash, the pellet was resuspended in 500 μL PBS.

For staining with the LIVE/DEAD Vio dye (LIVE/DEAD Fixable Violet Dead Cell Stain Kit, Thermo Fisher Scientific, Waltham, MA, USA, cat. no. L34955), the dye stock solution was prepared by diluting one vial of dye in 50 μL of DMSO. Subsequently, 1 μL of dye was used for staining of 500 μL of bacterial stock suspension. After incubation at room temperature in the dark for 1 h, labeled bacteria were washed three times with 500 μL of PBS using centrifugation for 2 min at 2350 *g*. Finally, the pellet was reconstituted in 200 μL of PBS. The labeled bacterial cells were counted and stored at 4 °C for up 24 h prior to use.

Additionally, for simultaneous confocal microscopy experiments, bacteria were conjugated with FITC and biotin as previously described [44]. The efficiency of conjugation was assessed by measuring fluorescence signal intensity of L/D Vio or FITC using BD LSRFortessa flow cytometerv (BD Biosciences, Franklin Lakes, NJ, USA), using 405 nm laser with a 450/50 nm filter for L/D Vio; 488 nm laser with a 530/30 nm filter for FITC; 561 nm laser with a 582/15 nm filter for AF568 and were compared with the control fixed, unconjugated bacteria. Next, L/D Vio (or FITC) labeled *E. coli* were conjugated with biotin using a FluoReporter Mini-biotin-XX Protein Labeling Kit (Thermo Fisher Scientific, MA, USA; cat. no. F6347) in accordance with the manufacturer’s instructions. Sterile glycerol was added to the labeled bacterial suspension to a concentration of 10%, and the *E. coli* concentration was adjusted to 10^10^ cells/mL with PBS/10% glycerol solution. The suspension was aliquoted into 20 µL stock, frozen, and stored at −20 °C for 2–3 months until use.

To evaluate biotinylation efficiency, 1 μL (1 µg) of streptavidin-AF568 (Thermo Fisher Scientific, MA, USA, cat.no. S11226) was added to 300 μL of a double-labeled bacterial suspension (10^7^ *E. coli*/mL). After incubation at +4 °C for 30 min, the bacteria were washed twice with PBS by centrifugation for 2 min at 2350 *g*, resuspended in 200 μL of PBS, and analyzed on a BD LSRFortessa to assess the percentage of bacteria positive for both L/D Vio and AF568 fluorescence.

### 2.2. Blood Sampling

Healthy donors from different age groups were selected for the study after providing voluntary informed consent. Qualified personnel at the medical center of Sirius University collected venous blood samples from the median cubital vein. The study was approved by the Ethics Committee of Sirius University of Science and Technology (protocol dated 6 March 2023). The list of the donors and the additional information about donors’ health are attached in Appendix A (see Appendix A).

Volunteer donors participating in the study were divided into two age groups: 14 donors (7 women) aged 22–33 years (junior) and 11 donors (6 women) aged 61–73 years (senior).

Qualified veterinarians from the National Research Center “Kurchatov Institute” Medical Primatology Complex selected a group of sexually mature, healthy cynomolgus monkeys (*Macaca fascicularis*), also known as crab-eating or long-tailed macaques. The macaques were divided into two age groups: 14 animals (5 females) aged 4–7 years (junior), corresponding to approximately 15–23 human years, and 7 animals (3 females) aged 19–27 years (senior), corresponding to approximately 62–89 human biological age.

All monkeys were originally imported from Southeast Asia and bred for a long time (several generations), at a breeding facility in Sochi.

Animals were housed individually or in family groups under ambient environmental conditions (temperature 21–28 °C, relative humidity 40–70%) with exposure to natural daylight. Sanitary measures were conducted in accordance with applicable local regulations. Animals had ad libitum access to fresh drinking water supplied via a centralized water system in compliance with state sanitary standards (SanPiN 2.1.3684-21, SanPiN 1.2.3685-21 [45,46]; Rospotrebnadzor, Moscow, Russia; https://www.rospotrebnadzor.ru/files/news/SP2.1.3684-21_territorii.pdf, assessed on 22 December 2025). Water quality was regularly monitored, and samples were periodically analyzed for potential contamination.

The animals received a nutritionally balanced diet containing adequate amounts of proteins, fats, carbohydrates, vitamins, and dietary fiber. The diet consisted of complete feed, rice porridge with raisins, vegetables, fruits, briquetted feed, and biscuits, provided in accordance with average daily feed intake standards. Feeding was performed three times daily: from 8 a.m. to 9 a.m. animals received granulated balanced feed; from 11 a.m. to 12 p.m. juicy feed (fruits and vegetables), cookies, and rice porridge; and from 2 p.m. to 3 p.m. granulated balanced feed, in accordance with local regulations (SOP AC 11 “Feeding Individually Caged Monkeys”).

All experimental procedures were conducted in accordance with the guidelines of the International Association for Evaluation and Accreditation of the Care of Laboratory Animals. All macaques included in the study were clinically healthy, as confirmed by routine veterinary examinations, negative test results for tuberculosis and herpes virus B, and a review of previous medical records.

After anesthesia, a qualified veterinarian collected blood from the superficial femoral vein and bone marrow from the trabecular zone of the tibia using a Jamshidi needle. The work was carried out in accordance with the approved ethical permission of the Kurchatov Complex of Medical Primatology (No. 02-3pr dated 21 March 2024).

All blood samples were collected in tubes with heparin to prevent coagulation (BD, Vacutainer), then the number of leukocytes, granulocytes, monocytes and lymphocytes was measured in whole blood using a MEK-7300K hemoanalyzer (Nihon Kohden, Tokyo, Japan). For erythrocyte lysis, 10 volumes of BD Pharm Lyse™ lysing buffer were added to the cell pellet and incubated for 10 min at room temperature. After incubation, the cells were resuspended in BD Biosciences Cell Wash containing antibodies.

Following the “3R” principles (Replacement, Reduction, and Refinement) that guide ethical and humane animal research, aiming to minimize the use and suffering of animals, some studies were not performed on experimental animals due to the lack of age differences in human myeloid cell populations.

### 2.3. Testing the Enzymatic Activity of SA-β-Gal in Living Blood Cells

After erythrocyte lysis, blood cells were resuspended in RPMI medium with 10% FBS and additional glutamine (final concentration—2 mM, all from Capricorn Scientific, Ebsdorfergrund, Germany) using centrifugation for 5 min at 330 *g*. Cells were placed in a CO_2_ incubator for 1–2 h (5% CO_2_, 37 °C), then counted, and cell integrity was controlled on an Evos M3000 fluorescence microscope (Thermo Fisher). Cells were resuspended in 250 μL of medium with bafilomycin A (final concentration 100 nM), incubated for 1 h (5% CO_2_, 37 °C). After completion of incubation, SPiDER reagent (Dojindo Molecular Technologies, Rockville, MD, USA; SG0403), 1 μM final, was added to the leukocytes for vital staining for SA-β-Gal activity and incubated for 1 h (5% CO_2_, 37 °C), according to the manufacturer’s instructions. Finally, the cells were washed by centrifugation and resuspended in a buffer suitable for further procedures.

### 2.4. Phagocytosis Test

The cells were washed by centrifugation once for 5 min at 330 *g* and resuspended in RPMI/10% FBS/Glutamine. An aliquot of the cells was stained 1:1 with acridine orange (final concentration 10 μg/mL) and the concentration of live cells was counted using Countess 3 (Thermo Fisher Scientific, MA, USA). Leukocytes at a concentration of 500,000 cells in 100 μL (100 μL/sample) were placed in a polypropylene non-adherent tube (Eppendorf, Hamburg, Germany), the temperature was stabilized in a CO_2_ incubator (37 °C, 5% CO_2_) for 5 min and the *E. coli* preparation was added (leukocytes: *E. coli* ratio 1:20). The *E. coli* suspension was prepared as described in Section 2.1. The suspension was incubated for 1 h at 37 °C, 5% CO_2_ and then placed on ice to stop phagocytosis. Next, 10 volumes of BD CellWash Buffer were added to the cells, centrifuged at 330 *g* for 5 min (10 °C), and the supernatant containing unabsorbed bacteria was removed. Staining with antibodies and streptavidin-DL550 (DyLight 550) or streptavidin-AF568 was performed to identify subpopulations and bacteria remaining outside on the cell surface.

### 2.5. Confirmation of Localization of Bacteria on Phagocytes

To validate the developed protocols, flow cytometry and confocal microscopy were simultaneously performed as described in our previous work [44]. Figure 1A presents a diagram illustrating the possible distribution of bacteria inside and on the surface of phagocytes based on their presumable fluorescence; Figure 1B shows a representative FACS plot depicting the fluorescence of blood cells following phagocytosis and streptavidin staining. Bacteria that were phagocytosed during a one-hour incubation fluoresced only in the FITC channel, indicating their internalization. In contrast, bacteria attached to the cell surface but not engulfed remained accessible for additional staining with streptavidin–DL550, allowing identification of their surface localization.

For comparison of the two methods, after phagocytosis and staining, double-positive cells (containing bacteria on the outside) were sorted using a BD FACSAria III. These sorted cells were mounted on glass slides for confocal microscopy analysis. Representative cells were imaged layer by layer (Z-stack with 1 µm intervals) at different positions along the *Z*-axis (Figure 1C). In this example, bacteria located on the cell surface appear in the upper layers and are double positive for FITC and streptavidin–DL550, whereas bacteria inside the cells are stained only with FITC and show no streptavidin–DL550 staining. Multiple engulfed bacteria within granulocytes are further illustrated in Appendix A.

### 2.6. Flow Cytometry and Cell Sorting

After discarding the supernatant containing unabsorbed bacteria, the blood cell suspension was for 30 min stained with set of antibodies (Appendix A) on ice, and then the cells were washed once by adding BD CellWash to 1.5 mL and centrifuging for 5 min at 330 *g*.

Samples were analyzed using BD LSRFortessa flow cytometer. The gating strategy involved sequential identification of all leukocytes based on forward scatter (FSC) and side scatter (SSC) parameters, followed by cell duplets exclusion. Subpopulations of phagocytic leukocytes were then identified according to fluorescence intensity signals as negative for CD3, CD19/20 and positive for CD14 or CD16 (Appendix A). These subpopulations were analyzed for phagocytosis by fluorescence of L/D Vio and streptavidin-AF568. The complete gating strategies are presented in Appendix A.

To perform confocal microscopy, myeloid blood cells before and after completion of phagocytosis, were sorted using BD FACSAria III for side scatter (SSC) and CD45-APC-eFluor780, into CD45^+^ population of phagocytes, as well as into a single positive and double positive population having *E. coli* inside or both outside and inside the phagocytes. Then the cells were used for additional antibody staining or directly for slides mounting for confocal microscopy.

### 2.7. Analysis of Genetic Instability Using HMGB1 and γH2AX Markers

3 mL of whole blood, after erythrocyte lysis, was resuspended in BD CellWash buffer. The cells were washed, and live, singlet cells were sorted based on side scatter (SSC) into monocyte and granulocyte populations using a BD FACSAria III flow sorter. After sorting, the cell populations were resuspended in PBS and then transferred to fixation and permeabilization buffers for intracellular staining, following the manufacturer’s protocol for the BD Cytofix/Cytoperm™ Fixation/Permeabilization Kit.

Briefly, the cells were washed with PBS by centrifugation (5 min at 330 *g*), and the cell pellet was immediately resuspended in 100 μL Cytofix/Cytoperm (BD, #51-2090KZ) and incubated for 20 min at 4 °C. The cells were centrifuged, the supernatant was discarded, and the pellet was resuspended and washed twice with 200 μL ready-to-use solution Perm/Wash buffer (BD, #51-2091KZ). Finally, the cells were resuspended in 50 μL Perm/Wash buffer containing either anti-γH2AX antibodies (polyclonal, Cusabio, Wuhan, China; cat. #CSB-PA105411, 1:100) or anti-HMGB1 antibodies (clone EPR3507, Abcam, Waltham, MA, USA; cat. #ab79823, 1:100). The cells were incubated at 4 °C for 16 h in the dark. After primary antibody incubation, the cells were washed twice with 200 μL ready-to-use solution BD Perm/Wash buffer for 5 min each. The cells were then resuspended in 100 μL Perm/Wash buffer containing goat anti-rabbit AF555 secondary antibody (polyclonal, Abcam, cat. #ab150086, 1:500) and incubated at room temperature for 1 h. The cells were washed 5 min twice with 200 μL Perm/Wash buffer. Next, the cells were stained with DAPI (1 μg/mL) for 5 min in the dark at room temperature, followed by two washes with 200 μL PBS for 5 min each. After staining, the cells were concentrated by centrifugation for 5 min at 330 *g*, resuspended in 10 μL PBS, and placed onto Polysine Adhesion Slides (Thermo Fisher Scientific, MA, USA). The slides were incubated in a humid chamber for 15 min at room temperature and then mounted in Fluoroshield medium (Abcam, cat. #ab104135) according to the manufacturer’s instructions. Finally, the samples were covered with coverslips and sealed with transparent nail polish.

### 2.8. Confocal Microscopy

Confocal fluorescence images were obtained using an inverted point scanning confocal microscope (LSM 980 based on Axio Observer 7, Carl Zeiss, Oberkochen, Germany) through an oil immersion 63× objective (Plan-Apochromat, NA 1.4, Carl Zeiss, Germany). We used simultaneous acquisition setup to obtain 3D images of macrophages and bacteria (Figure 1C) FITC, DL550 and CD45-APC-eFluor780 channels were acquired by using 488 nm, 543 nm and 639 nm lasers simultaneously; AOTF transmission was set to 0.0008%, 0.4% and 1.1%, respectively. Three channels of spectral detector were set as follows: 498–532 nm for FITC, 562–629 for DL550 and 649–750 for CD45-APC-eFluor780. Detector gains were adjusted to fill image dynamic range. Confocal scan zoom was set from 3× to 7× depending on cell size. The image parameters were set as follows: pixel dwell time 1.49 µs; pixel size dx = dy = 0.08 µm, z-stack step dz = 0.4 µm. To obtain images of Alexa Fluor 555 channel (HMGB1 and γH2AX, with good signal-to-noise ratio, Alexa Fluor 555 and DAPI channels were acquired sequentially by using 543 nm and 405 nm lasers, respectively; AOTF transmission was set to 0.1% and 0.2%, respectively. Spectral detector band was set to 545–641 for Alexa Fluor 555 and to 399–484 for DAPI. Detector gains were adjusted to fill image dynamic range on brightest samples. Confocal scan zoom was set from 7× to 9× depending on cell size. The image parameters were set as follows: pixel dwell time 4.95 µs; pixel size dx = dy = 0.07 µm, z-stack step dz = 0.19 µm. All images were obtained using Zen software (version Zen Blue 3.2, Carl Zeiss, Germany).

### 2.9. Programing Images Quantification

The results of HMGB1 and γH2AX staining were processed using Anaconda Navigator and Jupiter Notebook in Python v.3.10, using built-in libraries. The processing principle is presented below.

To visualize subcellular (throughout entire the cell volume) the distribution of fluorophores, three-dimensional confocal image stacks were converted into two-dimensional intensity maps. This was achieved by summing voxel intensities along the optical (z) axis for each fluorescence channel, producing a single projection image per cell. This approach effectively transforms volumetric data into a representative 2D image, analogous to the median fluorescence intensity (MFI) metric used in flow cytometry, capturing the total fluorophore content on a pixel-by-pixel basis

The resulting projected images were then linearly rescaled to utilize the full 8-bit range, enhancing contrast and compensating for variability in absolute signal between cells. An automatic global threshold was determined using Otsu’s method to distinguish cellular regions from background noise. The largest contiguous object identified in the binary mask was designated as the cell of interest.

To differentiate nuclear versus cytoplasmic compartments, a secondary channel mask was applied (Figure 2). Specifically, subtracting the inverse binary mask isolated cytoplasmic pixels, while subtracting its complement isolated nuclear pixels. For each compartment, pixel intensities were compiled into histograms representing the full fluorescence distribution across all pixels within that region. The half-width at half-maximum (HWHM) of each histogram served as a quantitative measure of distribution breadth, reflecting fluorophore dispersion within the compartment.

Within each segmented region, two key metrics were calculated:N: The total number of non-zero pixels (i.e., pixels with detectable fluorescence).S: The sum of all pixel intensities (integrated fluorescence signal).Dividing S by N yielded the mean pixel intensity, serving as an image-based analog to flow cytometry’s MFI.

To compare distribution widths between nuclear and cytoplasmic compartments (and across experimental conditions), HWHM values were statistically analyzed using a one-way ANOVA, followed by Tukey’s HSD test (significance threshold *p* < 0.05). This statistical framework allowed for rigorous assessment of differences in fluorophore dispersion within cellular compartments, extending the MFI concept to spatially resolved fluorescence imaging.

Finally, pixel value histograms for each compartment were constructed as*H*(*k*) = #{(*y*,*x*)|I*_sub_*(*y*,*x*) = *k*},*k* ∈ [1, 255]
where I_sub_(y, x) is the intensity of the subcellular region at coordinate (y, x).

The data processing program is available at the following reference link: https://github.com/ElijahBiocinth/MFI-h2ax (accessed on 31 October 2025).

### 2.10. Determination of Proinflammatory Cytokines Production

Whole blood was drawn into vacuum tubes containing clot activator and serum separation gel (1 mL). The samples were allowed to clot, centrifuged to separate serum, which was then aliquoted, frozen, and stored at −80 °C for subsequent cytokine analysis.

For assessment of myeloid cell cytokine secretion after stimulation the whole blood samples were aliquoted into 48-well culture plates (NEST, Wuxi, China) at 250 µL per well. The plates were incubated at 37 °C with 5% CO_2_ for 2 h to allow cell adherence. After incubation, non-adherent cells were discarded and then washed twice with phosphate-buffered saline (PBS). Adherent cells were then cultured in RPMI 1640 medium supplemented with 10% fetal calf serum (FCS), glutamine, and HEPES buffer. Cells were stimulated with either purified *Escherichia coli* lipopolysaccharide (LPS; final concentration 100 ng/mL, Sigma-Aldrich, St. Louis, MO, USA) or with fixed *E. coli* bacteria at a leukocyte-to-bacteria ratio of 1:20. Supernatants were collected 22 h post-stimulation, centrifuged to remove cell debris, and stored at −80 °C until use. Cytokine levels (TNF and IL-6) in the supernatants were analyzed using enzyme-linked immunosorbent assay (ELISA) kits (Vector-Best, Koltsovo, Novosibirsk region, Russia), following the manufacturer’s instructions. Supernatants from unstimulated cultures served as negative controls.

### 2.11. Analysis of Antibacterial Properties of Plasma and Whole Blood

The day before the experiment, to obtain an overnight bacterial culture, 50 μL of *E. coli* stock (strain *DH5α*) in 10% glycerol was placed in 10 mL of LB medium and incubated at 37 °C with shaking at 200 rpm in Excella E25R shaker-incubator (New Brunswick Scientific, Edison, NJ, USA). The next day, the optical density of the bacterial suspension was measured at 600 nm using a Multiskan SkyHigh microplate spectrophotometer (Thermo Fisher Scientific, MA, USA), as described above. The overnight culture was divided into two parts: one part was fixed as described above, and the other part was kept alive. The concentration of fixed bacteria was determined using a Novocyte 3000 flow cytometer (Agilent, Santa Clara, CA, USA), based on forward and side scatter indices. The unfixed (alive) portion of the overnight *E. coli* culture was serially diluted for further experiments.

To obtain plasma, 2 mL of whole heparinized blood from each donor was centrifuged for 7 min at 600 *g*. The hematocrit of each donor was measured using a hematology analyzer MEK-7300K (Nihon Kohden, Tokyo, Japan) to determine the plasma volume equivalent to 100 μL of whole blood. The plasma volume equivalent to 100 μL of blood for each donor was calculated using the formula: plasma volume = 100 μL of whole blood—hematocrit. To determine the titer of the initial stock, bacteria were plated on LB-agar plates using limiting dilution concentration. The number of bacterial colony-forming units (CFU-B) was counted after 24 h. This titer value was taken as the control concentration of bacteria at zero-time point (C_0_). Bacteria were then incubated in 100 μL of whole blood and an equivalent volume of plasma in Eppendorf tubes on a rotary shaker (60 *g*) at 37 °C. The first group of whole blood and plasma samples was pre-incubated with 10^8^ fixed bacteria for 4 h, after which live bacteria (10^7^ per sample) were added and incubated for an additional 4 or 24 h. In the second group, live bacteria (10^7^ per sample) were added immediately to the sample and incubated for 4 h.

To assess the effect of plasma on bacterial viability in the absence of antibacterial and humoral factors, plasma was heat-inactivated at 56 °C for 30 min, immediately cooled, and then incubated with live bacteria for 4 h. After incubation, all samples were plated on LB agar Petri dishes and incubated overnight at 37 °C. Colony counting was performed after 24 h.

To evaluate the neutralizing capacity of the samples, for each sample, the CFU-B values after incubation of plasma with bacteria were normalized to the control CFU-B at the zero-time point (C_0_), calculated at the beginning of the experiment before incubation with blood components.

### 2.12. Software and Statistical Analysis

Flow cytometry data were analyzed with BD FlowJo v. 10.10. Confocal microscopy images were processed with ImageJ/Fiji v. 1.54f.

Primary flow cytometry data, phagocytic assay results, were documented, processed, and composed in Microsoft office 2019. Data visualization and statistical analyses were performed using GraphPad Prism v.9.5.0, utilizing its built-in statistical tools. Statistical significance is indicated in the graphs as follows: *p* < 0.05 (*), *p* < 0.01 (**), *p* < 0.001 (***), *p* < 0.0001 (****), and ns (not significant). During study, researchers were “blinded” to the age group and sex of the samples during data acquisition and analysis.

### 2.13. Limitations of the Study

Aging, including aging of the immune system, is a non-linear process [47]. Systemic changes in multiple molecular components associated with aging occur nonlinearly, with increases observed between the ages of 40 and 60 years [48]. Additionally, the average age at diagnosis of an autoimmune disease is 54 years [49]. In our study, we demonstrated the stability of innate immune system cells with increasing age between younger (22–33 years) and older (57–73 years) groups. As we interested in physiological aging, the study only included healthy working adults and did not encompass individuals with compromised immune function, the very elderly (over 80 years), or middle-aged individuals (35 to 55 years).

In contrast to studies based on clinical cohorts, the primary aim of the present study was to characterize universal indicators of innate immune system aging typical of an average, clinically healthy individual regardless of sex, rather than to identify individuals with accelerated aging or early-onset pathologies within the population. Furthermore, one of the primary objectives of this study was to compare aging-associated evolutionally conserved traits between macaques and humans.

Although the study included a cohort of elderly macaques, many of which were of near-terminal age, the sample did not display pronounced signs of innate immune system aging. It should be noted that animals surviving to such advanced ages represented a group of clinically healthy dominant males and females who may have benefited from superior access to resources due to their privileged social status within the group. In this context, the study can be viewed as an investigation of longevity in non-human primates, potentially associated with a robust innate immune system. Additionally, due to restricted access to the samples and labor-intensive experiment designs, we examined, in general, 25 humans (11 senior) and 21 monkeys (7 senior) which is a sample size slightly exceeding the minimum required for statistical analysis. Thus, the study is limited to identifying moderate changes that may occur during the aging process.

In addition, we assessed the stability of innate immune cells by analyzing selected markers of senescent cells. These markers include γH2AX, HMGB1, and SA-β-Gal. These markers may indicate cellular stress, genetic instability, and impaired lysosomal function. We also compared the phenotype, secretion profile (TNF and IL-6), serum and blood antimicrobial properties and functional properties of cells from groups of young and elderly donors. However, there are other markers of aging, such as ROS, telomere shortening, epigenetic drift, and mitochondrial dysfunction. These markers have not been studied and may be of interest for future research. However, cellular functions remained stable across age groups, suggesting that the aforementioned molecular processes were not disrupted in this case.

In conclusion, we found a predominance of humoral factors over cellular factors in antimicrobial defense (against *E. coli*, strain DH5α). The uniqueness of our approach is in the simultaneous assessment of both cellular and humoral blood functions, which enables a better understanding of how the immune system combats various bacteria—specifically, whether they are sensitive or insensitive to complement. Thus, the developed protocols can be used for any strain to comprehensively understand the role of cellular and humoral factors in various infectious diseases and infectious immunity in general.

## 3. Results

### 3.1. Stability of Immunophenotype Across a Broad Age Range in Human and M. fascicularis

To determine age-related differences in immune cell content in blood and bone marrow samples from humans and macaques in two age groups (junior and senior, see Section 2), the percentages of major cell populations were analyzed using flow cytometry. However, no statistically significant differences in the composition of the main blood cell populations were found between age groups of donors (Figure 3A). Only a tendency toward a decrease in the proportion of granulocytes in the blood was observed in elderly individuals of both species. However, in the bone marrow of macaques, the granulocyte percentage, conversely, tended to increase with age. To study interspecies differences, three groups of samples were compared: human blood and the blood and bone marrow of cynomolgus macaques (Figure 3B). The number of immune cells in human and macaque blood samples did not differ statistically, while statistically significantly fewer T- and B-lymphocytes were found in the bone marrow than in the blood samples, and more granulocytes were found in the bone marrow. A trend toward a decrease in the number of non-classical CD16^+^CD14^lo^ monocytes (non-classical monocytes, ncMOs) and CD14^+^CD16^+^ monocytes (intermediate monocytes, inMOs) was also detected in macaque bone marrow.

Thus, the results indicate that peripheral blood and bone marrow differ in the relative proportions of lymphocytes and granulocytes, with the latter showing an age-associated tendency to accumulate in the bone marrow. However, in the study population, no significant differences in the proportion of myeloid populations were observed between elderly and young individuals in either humans or monkeys.

### 3.2. Quantification of Genomic Stability in Human Myeloid Cells Across a Broad Age Range

As described previously, cellular aging is accompanied by an increase in genetic instability [50]. In this study, we developed a method for quantifying nuclear and cytoplasmic fluorescence intensities by applying dimensionality reduction to a numerical parameter used in flow cytometry—mean fluorescence intensity (MFI)—to analyze staining intensity at the single-cell level. This approach enabled quantifying the staining of myeloid blood cells with antibodies targeting HMGB1 and γH2AX, allowing separate assessment of nuclear and cytoplasmic staining for each marker (Figure 4). Specifically, for HMGB1, we evaluated its content in the nucleus and cytoplasm, while for γH2AX, we assessed its nuclear content relative to cytoplasmic background. Although γH2AX can also be detected in the cytoplasm during early apoptosis [51], apoptotic cells were not observed in our samples.

Cells from donors in two age groups were stained to compare markers of genetic instability associated with cellular aging. We hypothesized that senescent myeloid cells from elderly donors would exhibit differences in these markers. The analysis included cells from both junior and senior donors. For each donor and marker, thirty granulocytes and monocytes sorted from blood were analyzed. Images were processed separately to evaluate the distribution of HMGB1 and γH2AX staining within the nucleus and cytoplasm. Nuclear areas were defined based on DAPI staining (Figure 4).

Comparison between the two healthy donor age groups revealed no statistically significant differences in fluorescence intensity or in the localization (nuclear versus cytoplasmic) of HMGB1 and γH2AX staining (Figure 4B).

### 3.3. High SA-β-Gal Activity in Myeloid Cells Remains Stable During Aging in Two Species

Another important marker of cellular aging is the increased activity of the senescence-associated β-galactosidase (SA-β-Gal) enzyme. Recently, a highly efficient protocol for detecting SA-β-Gal activity has been developed. This method utilizes the SPiDER dye, which is effectively immobilized within cells, allowing for accumulation of a fluorescent signal and increased sensitivity to β-galactosidase activity. While the optimal enzymatic activity of general β-galactosidase in young cells occurs at pH 4, the activity of SA-β-Gal in senescent cells peaks around pH 6. To facilitate this, the pH was shifted to the optimal range for SA-β-Gal activity using the pH modulator Bafilomycin A1 (Baf A1), thereby preferentially enhancing SA-β-Gal activity within the cells. Under these conditions, increased SA-β-Gal activity can be observed in aged cells [52,53]. Additionally, both we and other researchers have demonstrated that this dye can be used in conjunction with antibodies in multicolor flow cytometry, enabling the isolation of cell populations with high SA-β-Gal activity [8,54,55].

A study of SA-β-Gal activity in human blood cell samples from two age groups, as well as in monkey blood and bone marrow cells, showed no increase in SA-β-Gal activity with age within myeloid populations and B cells in either species (Figure 5A). Moreover, in bone marrow samples from elderly monkeys, a trend toward decreased SA-β-Gal activity was observed.

However, significant differences were detected between cell populations regardless of age. For example, granulocytes and monocytes—particularly the CD14^+^CD16^lo^ monocyte subset (classical monocytes, cMOs)—exhibited high SA-β-Gal activity in blood cells (Figure 5B). Moreover, SA-β-Gal activity was significantly higher across all myeloid populations compared to lymphoid populations (B cells) in both species. This finding aligns with data from rodents, where both endogenous β-Gal and SA-β-Gal activity were markedly elevated in myeloid cells [55].

A comparative analysis of SA-β-Gal activity in immune cell populations from the blood and bone marrow of macaques showed that SA-β-Gal activity is significantly higher in the blood—particularly in the major myeloid populations, including granulocytes and cMOs—compared to the bone marrow (Figure 5C). Since total β-Gal level measured here reflects increased lysosomal activity (lysosomal mass) and perhaps of carbohydrate metabolism within lysosomes, it can be inferred that these myeloid populations become activated upon entering the peripheral circulation. This is especially evident in cMOs, whereas ncMOs exhibit increased SA-β-Gal activity specifically in the bone marrow. These differences may indicate distinct functional roles for these populations in peripheral blood versus bone marrow.

### 3.4. Phagocytosis Function and Proinflammatory Cytokine Secretion Remain Stable Across a Wide Age Range, Demonstrating Evolutionarily Conserved Features of Myeloid Cells in Both Humans and M. fascicularis

To investigate age-related changes in the functional activity of myeloid cells, we examined one of the key aspects of innate immunity—the phagocytic activity of monocytes and granulocytes. Cell phagocytic activity was assessed in whole blood or in RBC-lysed blood without prior adhesion. Phenotypic analysis of granulocytes—the main cell type involved in phagocytosis—revealed significant differences in phenotype between humans and cynomolgus monkeys. Human granulocytes lack surface CD14, a component of the CD14/TLR4/MD2 receptor complex that recognizes LPS. In contrast, macaque blood contains an average of 80.4 ± 24.1% CD14^+^ granulocytes, and macaque bone marrow contains 96.9 ± 13.3% CD14^+^ granulocytes. Additionally, macaque granulocytes were negative for the CD16 (FcγRIII) marker, whereas in humans, approximately 92.4 ± 13.5% of granulocytes were CD16^+^ (Figure 6A,B). Despite these phenotypic alterations, no significant differences were observed in the phagocytic functional properties of macaque and human granulocytes (Figure 6C,D).

The in-house developed fixed bacterial prep carried two labels to distinguish between engulfed bacteria and those unengulfed but trapped on the surface. This enabled the determination of the spatial localization of bacteria, distinguishing those inside (in) and outside (out) the phagocytes (see Section 2). In this context, age differences in phagocytic activity were also investigated. As a result of the analysis, no age differences in phagocytosis were found in myeloid populations over a wide range of ages in human and macaque blood samples, and macaque bone marrow (Figure 6C). Granulocytes are the most active bacterial scavengers, given their prominent role in phagocytosis and their abundance in blood and bone marrow, thereby executing the majority of phagocytic activity in these compartments. However, when comparing monocytic subsets across all samples, it was observed that, in both humans and macaques, the cMOs tend to retain larger quantities of unengulfed bacteria on their surface during phagocytosis, compared to the ncMOs and granulocytes (Figure 6D) after a 1 h incubation. No other populations exhibited such a significant bias toward bacterial surface capture and retention. Moreover, this phenomenon was consistent across both species, suggesting that it is an evolutionarily conserved process.

Analysis of serum levels of the proinflammatory factors TNF and IL-6 showed no significant age-related differences in either humans or monkeys. Only a trend toward decreased IL-6 levels was observed in humans (Figure 6E). The response to stimulation with either LPS or whole bacteria also did not change significantly between the two age groups (Figure 6F).

To clarify the fate of unengulfed bacteria remaining on the surface of cMOs and granulocytes (GRs), we extended the incubation time of monocytes in suspension to 1, 2, and 4 h to assess the phagocytosis time course (Figure 7A). Results showed that the percentage of cMOs containing only internalized bacteria increased significantly by 4 h of incubation. Concurrently, there was a visible trend toward a decrease in the number of cMOs retaining bacteria on their surface. Since excess bacteria were added (leukocyte:bacteria ratio of 1:20), we hypothesize that cMOs can bind more bacteria than they can phagocytize, which may prevent bacterial dissemination throughout the body. This suggests that classical monocytes may undergo multiple rounds of phagocytosis lasting more than 4 h. In contrast, granulocytes showed an opposite trend: a decrease in cells containing only internalized bacteria and an increase in cells with bacteria attached to their surface.

Our data above showed that human granulocytes are negative for the CD14 marker. However, with prolonged incubation with bacteria, the MFI for CD14 significantly increases on granulocytes (by 4 h of incubation), although the overall CD14 level remains low compared to monocytes. Moreover, the median CD14 level in cMO decreases over 4 h, which is consistent with literature data on CD14 internalization into cells upon LPS stimulation (Figure 7B) [56].

### 3.5. A Dominant Contribution of Humoral Factors in the Immediate Control of Bacterial Infection

To compare the contributions of myeloid cells and plasma humoral factors to bacterial infection control across donors of different age groups, we developed an assay to assess bacterial viability after exposure to whole blood or plasma (Figure 8A). In all assays, we observed a significant reduction in colony-forming units of live bacteria (CFU-B) incubated with whole blood or plasma compared to the control (an aliquot of bacteria not exposed to blood components). However, we did not find statistically significant differences in CFU-B counts after exposure to whole blood or plasma from donors in the two age groups (Appendix A).

The comparative analysis of incubation with whole blood and plasma allowed us to evaluate the combined contribution of myeloid cell functions (degranulation, NETosis, phagocytosis) and plasma humoral factors alone. The results were assessed by measuring the reduction in viable bacterial colonies (CFU-B) after 4 h of incubation. The experiments employed limiting dilution, with live bacteria added at varying concentrations to generate results suitable for visual counting.

Results were compared against a control containing an average of 10^8^ CFU-B of *E. coli*. Variants 1 and 3 represent the results of bacteria exposed to whole blood, while variants 2 and 4 represent exposure to blood plasma (Figure 8A). In variants 1 and 2, cells were pre-incubated with paraformaldehyde-fixed bacteria for 4 h to deplete serum humoral factors and pre-activate myeloid cells. The control consisted of CFU-B after 4 h of bacterial incubation in LB medium. Several experiments demonstrated that preincubation of blood plasma at 56 °C for 30 min, followed by a 4 h incubation with bacteria, did not significantly alter the number of CFU-B compared to incubation in LB medium alone, which is optimal for supporting *E. coli* growth. This indicates that native humoral protein factors are essential for bacterial neutralization, and thermal inactivation effectively abolishes the bactericidal activity of blood plasma. Figure 8B presents the statistical analysis of the data from these experimental variants. Incubation with whole blood reduced the number of viable bacteria by approximately 6 orders of magnitude after 4 h, while incubation with plasma reduced viable bacteria by about 6–7 orders of magnitude on average. This suggests that the volume of plasma, equivalent to 100 µL of blood used, can kill approximately 10^7^ *E. coli* bacteria, highlighting the high bactericidal capacity of plasma alone.

Statistical analysis revealed significant differences between plasma samples pre-incubated with killed bacteria (variant 2) and those without such pre-incubation (variant 4). This indicates that plasma factors, such as components of the complement system and antibodies, interact with fixed bacteria, thereby depleting the plasma’s bactericidal potential (Figure 8B).

No statistically significant differences were observed when comparing the contributions of plasma alone versus whole blood (naturally containing plasma and cellular components) to bacterial killing.

Conversely, a tendency toward reduced neutralizing capacity was noted in whole blood samples (variant 3) compared to plasma samples (variant 4). Therefore, the data do not establish a significant role for the functional activity of innate immune cells in the neutralization of the Gram-negative *E. coli* strain *DH5α*. Nonetheless, the protocol we developed can be adapted to investigate the contributions of cellular blood components in the clearance of other pathogens.

## 4. Discussion

Analyzing cell function in whole blood or in suspension ex vivo enable to approximate physiological conditions in vivo and model the early stages of blood bacterial infection. As previously demonstrated, neutrophils are capable of phagocytosing in suspension without prior adhesion [57]. In this study, granulocytes (neutrophils) and monocytes from the whole blood of humans and cynomolgus monkeys were incubated with bacteria on non-adhesive plastic surfaces with stirring in a rotary mixer. These experiments suggest that, upon entering the bloodstream, an opsonized pathogen can be effectively eliminated by most phagocytes in suspension, without the need for substrate attachment.

A group of donors representing the most productive, working age was selected for the study. The ages of these donors ranged from 22 to 73 years. The entire donor cohort was generally healthy and free of any chronic or acute diseases. Additionally, a group of non-anthropoid primates (*Macaca fascicularis*), with biological ages similar to those of the human participants, was investigated. The elderly macaques of this age range had not been previously studied and serve as a unique model for exploring evolutionarily conserved mechanisms of primate aging. The study objectives included analyzing quantitative changes in cell populations, functional activity, and aging markers within myeloid populations of both humans and non-anthropoid primates. We hypothesize that identifying early markers of immune system aging may help in finding ways to prevent premature aging and slow the rate at which people enter the zone of exponential growth of pathological processes.

Current understanding of immune system aging highlights alterations in the cellular composition and functional capacity of the adaptive immune system, along with increased genetic instability and accumulation of senescent cells in tissues, commonly detected by elevated activity of the SA-β-Gal enzyme. Additionally, a hallmark of aging is a myeloid bias within the immune system, characterized by a shift in blood stem and progenitor cells development toward myeloid lineages, resulting in an increased accumulation of myeloid cells in the bloodstream [3,4,5,58,59].

However, it remains unclear whether myeloid cells lose their basic functions and what causes their accumulation. Is this a consequence of quantitative compensation due to a decrease in the functional activity of the myeloid cells themselves, or is this quantitative compensation caused by a decrease in the activity of the adaptive lymphoid part of the immune system, while the monocytes themselves retain their non-attenuated function? In other words, do myeloid cells retain their “naïve” functions, or do they transition into a different, potentially pathological or dysfunctional state that may underlie immune aging? Additionally, does this aging process of the innate immune system begin during the productive, pre-retirement age when individuals typically feel relatively healthy? Finally, are these features of immune aging conserved among primates?

In the first stage of the study, the quantitative dynamics of myeloid cells across different age groups were analyzed. According to literature data, a shift in the classical or non-classical monocyte population may be observed in oncological, infectious and metabolic diseases [60,61]. In our study, we examined clinically healthy people of different age groups. As a result, no significant changes in the composition of myeloid populations, including subpopulations of monocytes, with age were detected. Based on the data obtained, it was concluded that neither the cell composition nor the expression levels of examined markers in myeloid population in humans and macaques change significantly over an examined age range.

The next stage of the study analyzed additional markers of cellular aging, including indicators of enhanced DNA repair and genetic instability. This was evidenced by increased nuclear expression of γH2AX and HMGB1, as well as their spatial localization within the cell. During cell aging or apoptosis, these markers change their localization, with increased abundance in the cytoplasm, providing information about the state of the cell’s genetic apparatus. Additionally, aging is one of risk factors for Clonal Hematopoiesis of Indeterminate Potential (CHIP) [62]. CHIP is characterized by specific mutations in hematopoietic stem cells compartment and hence their expansion. Aging is a risk factor for CHIP development due to the accumulation of somatic mutations in hematopoietic stem cells. By analyzing myeloid cells from donors of junior and senior groups, we expected to observe differences that would be due to differences at the level of early precursors of these cells. Previous studies of the spatial distribution of these markers in animal models and human cells have demonstrated a connection between age-related changes, genetic instability, cellular senescence, and the localization of γH2AX and HMGB1 [19,63].

To accurately characterize the expression of these markers of genetic instability, quantitative assessment of staining is necessary to measure the fluorescence intensity of these specific markers spatially and to statistically evaluate their distribution within the cell’s volume [64]. Confocal microscopy with subsequent spatial quantification did not reveal a statistically significant increase in cytoplasmic HMGB1 or an increase in nuclear γH2AX foci in myeloid cells from older donors.

Another indicator of cellular aging and the accumulation of senescent cells in the population is SA-β-Gal, an enzyme associated with aging. β-Gal is a product of the GLB1 gene [65]. During aging, the enzyme undergoes significant modification. For example, the optimal pH for enzyme function changes. Therefore, β-Gal activity at pH > 6 is considered abnormal and associated with aging. SA-β-Gal activity is associated with the accumulation of senescent cells with age. SA-β-Gal activity, which has an optimal activity at neutral pH, is associated with perturbation of lysosomes—an increase in their number due to stagnation of cellular processes within the cell and metabolic disturbances [66]. Although SA-β-Gal is considered a generally accepted marker for detecting senescent cells, some studies indicate that SA-β-Gal activity may even decrease in some age-related pathologies, such as type 2 diabetes [67]. Also, the determination of SA-β-Gal activity may vary depending on the research method. Thus, in a study on donor blood monocytes, where SA-β-Gal detection was carried out on fixed samples, an increase in enzyme activity in adhered monocytes with age was found [35]. Another study demonstrated that changes in SA-β-Gal levels in mouse cells—either increases or decreases—depend on the type of cell stimulation, regardless of age. During macrophage differentiation, SA-β-Gal activity decreased in response to M1 stimuli (such as LPS and IFN-α), while it increased in response to M2 stimuli (IL-4 and IL-13). The study concluded that SA-β-Gal activity may be linked to the immunological function of macrophages, and a high level of expression is considered a normal physiological characteristic of these cells [68]. SA-β-Gal exhibited consistently high activity levels across myeloid populations over a wide age range, contradicting our initial hypothesis that SA-β-Gal serves as a marker of myeloid cell senescence and that its activity would increase with age. Simultaneous studies in two species revealed significant variation in SA-β-Gal activity among different cell populations in blood and bone marrow, but no age-related differences were observed.

The study presented here revealed significantly higher SA-β-Gal activity in myeloid blood cell populations (granulocytes and monocytes) compared to lymphocytes. Moreover, granulocytes and cMOs in the blood of both species had the highest SA-β-Gal activity. However, SA-β-Gal activity was reduced in the same populations in the bone marrow, while SA-β-Gal activity was highest in the ncMOs bone marrow population. Thus, despite the established notion that the accumulation of senescent cells with high SA-β-Gal activity is one of the default mechanisms of aging, our studies did not reveal such a trend in myeloid cells in either the blood or bone marrow of both species. Of course, accumulation of SA-β-Gal is associated with its abnormal activity related to aging. However, SA-β-Gal activity also correlates with increased endogenius β-Galactosidase activity, which reflects an overall increase in lysosome mass and intracellular catabolism. It is important to note that high levels of catabolism are especially characteristic of myeloid cells, which are specialized in disposing of cellular debris, performing phagocytosis, and clearing apoptotic cells (efferocytosis) [69].

Thus, in our study, myeloid populations in primates display elevated SA-β-Gal activity, which seems to be unrelated to age-related changes and instead reflects the metabolic characteristics of myeloid cells in blood and bone marrow. A similar pattern observed in rodents suggests the presence of an evolutionarily conserved mechanism, likely inherent to mammals [55].

The results of this study allow us to conclude that myeloid cells and their genetic integrity remain remarkably stable within this cohort. It is important to note that the purpose of the study was to analyze blood samples from employed humans. Therefore, the participants were actively working individuals without chronic diseases, generally in good health. This could have influenced the findings, as most studies associate immune aging with age-related diseases. Similarly, the animal study involved healthy animals under constant veterinary oversight, which may have contributed to the observed myeloid cells stability [66,70].

The classical view of blood monocytes is that they can directly participate in the innate immune response and, upon differentiation, become tissue macrophages performing phagocytic functions. In this study, all phagocytosis assays were conducted on whole blood cell suspensions to evaluate the ability of monocytes to control circulating bacteria. It is also known that cMOs persist in significant numbers in healthy adults and are involved in tissue surveillance and inflammatory responses. ncMOs are reported to exhibit the highest phagocytic activity, along with capabilities for antigen detection, chemotaxis, and uptake—functions critical to immune defense. inMOs, characterized by CD14^+^CD16^+^ expression, represent a transitional population involved in antigen presentation and the initiation of inflammation [71].

To assess cellular phagocytic activity, we employed a phagocytosis assay that analyzed the localization of bacteria relative to phagocytes surface using double staining. This approach allowed us to distinguish between cells containing bacteria internally and those with bacteria positioned outside (on the surface of the cells). After a 1 h incubation, the granulocyte and ncMO populations contained both bacteria-internalized (ncMO *E. coli* In) and bacteria located on the surface (ncMO *E. coli* Out) in approximately equal proportions. In contrast, the majority of cMOs—up to 99.1% in human blood, up to 92.0% in macaque blood, and up to 86.7% in macaque bone marrow—had bacteria predominantly retained on their surface.

To investigate the fate of bacteria captured by cMOs, incubation with bacteria was extended to 4 h. The results showed a statistically significant increase in the percentage of cMOs that had completely internalized bacteria (no bacteria on the surface, cMOs *E. coli* In). Concurrently, the proportion of cMOs with bacteria remaining on the surface tended to decrease after 4 h of incubation. Despite this, a substantial number of cMOs still retained surface-bound bacteria. Based on these findings, we hypothesize that cMOs prevent bacterial dissemination by retaining bacteria on their surface and conducting multiple rounds of phagocytosis.

Based on the data obtained in this study, bacterial retention on the surface of cMOs may represent a form of delayed phagocytosis, as cMOs are less capable than granulocytes of ingesting large quantities of bacteria. Through this retention mechanism, cMOs can digest and clear the initial wave of ingested bacteria before internalizing bacteria captured on their surface. According to the literature, cMOs also possess an increased ability to migrate to infection sites in response to the chemokine CCL2 and can differentiate into dendritic cells. Prolonged surface retention of bacteria may also be related to their potential migration and transport of bacterial debris to secondary lymphoid organs, where antigen presentation to naïve B cells can occur [72,73,74].

Surprisingly, our studies did not detect any age-related changes in either species in phagocytic activity or in bacterial surface retention within monocyte and granulocyte populations.

Further analysis also revealed significant species differences in granulocyte markers. In humans, the vast majority of granulocytes were positive for CD16 and negative for CD14. In *M. fascicularis*, on the other hand, granulocytes were CD14^+^CD16^−^. These two molecules significantly influence the type of cell function. For example, CD16 (FcγRIII) functions as a low-affinity receptor for the Fc fragment of IgG antibodies and plays a key role in triggering immune responses such as antibody-dependent cell-mediated cytotoxicity (ADCC), in which immune cells destroy antibody-coated target cells. CD14 is part of the TLR4 receptor complex, which recognizes bacterial LPS. These data suggest differences in preferred type of granulocytes activation upon encountering a pathogen between macaques and humans. Thus, it appears that the primary activator of granulocytes against bacteria in macaques is predominantly LPS, while in humans, it is opsonized fragments of the pathogen. However, when incubation with bacteria was extended to 4 h, an increase in CD14 on the surface of human granulocytes was detected. Thus, prior exposure to bacteria is necessary for CD14 upregulation and LPS recognition by human granulocytes. Despite phenotypic differences, studies of granulocyte function revealed no significant differences in the level of phagocytic activity in response to bacterial exposure in both macaque species and humans.

The knowledge gained about the differences in macaque and human granulocyte phenotypes may be useful for ADCC-preclinical testing in macaques and for studies using CD16 or CD14 targeting.

One of the manifestations of aging is the accumulation of senescent cells in the body that secrete inflammatory factors—components of the senescence-associated secretory phenotype (SASP). As described in the literature, this increases the overall inflammatory environment in the body, while the response of myeloid cells to pathogen stimulation is weakened.

A study using various subjects revealed a decrease in the cytokine response of blood populations in both mice and humans. Although the levels of the proinflammatory key factors IL6 and TNF remained unchanged in mouse plasma, after LPS stimulation of macrophages, significantly lower secretion of IL6 and TNF was observed in aged mice [75].

In another study, monocytes from elderly donors were found to have a significantly reduced monocyte response to TLR1/2 stimulation using the synthetic bacterial lipopeptide pam3Cys [76].

In our study, analysis of serum cytokine levels in humans and macaques across two age groups revealed no significant age-related differences in IL-6 and TNF levels. Furthermore, following stimulation with LPS or *E. coli* bacteria, there was no significant decrease in the responsiveness of blood PBMCs’ adhesion fraction to TNF and IL-6 in elderly donors; only a slight trend toward reduced response was observed. Consequently, myeloid cell functions, including phagocytosis and cytokine signaling, appear to remain unchanged within the age range studied. Additionally, no increase in key components of the SASP, such as TNF and IL-6, was detected in the serum of either macaques or humans.

The cellular response of the innate immune system (phagocytosis, NETosis, release of cytotoxic granules) is an important function of the innate immune system. It is also believed that phagocytes, through their phagocytic function, directly control infections by capturing and consuming large numbers of bacteria [77].

In this study, the contribution of innate immune cell function (including phagocytic function) was compared with the contribution of serum-dissolved factors (humoral factors) to the direct destruction of bacterial infections. The activity of humoral factors against pathogens is mediated by antimicrobial peptides—defensins and cathelicidin—that destroy the membranes of bacteria, fungi, parasites, and viruses; complement system proteins that can directly kill bacteria; antibodies that opsonize bacteria and activate complement; and plasma-dissolved albumin, which has antibacterial properties that destroy bacterial walls and biofilms [78]. Opsonization of bacteria facilitates elimination by phagocytes. Furthermore, antibodies activate the complement system, which can directly lyse bacteria, further enhance opsonization, and attract other immune cells to the site of infection by inflammatory cytokine expression. The production and activity of these humoral components of plasma are increased during an immune response

When comparing the contributions of cellular functions (including phagocytosis, degranulation, and NETosis) and plasma humoral factors to the control of *E. coli* infection ex vivo, it was found that cellular antibacterial mechanisms contributed minimally to bacterial killing compared to the potent bactericidal effect of plasma. Incubation with plasma resulted in a seven-order-of-magnitude (10^7^) reduction in viable bacterial colony-forming units, an effect that was entirely abolished after heat inactivation of the plasma. This confirms the crucial role of complement system in control of bacterial infections. In contrast, whole blood exhibited bactericidal activity that was comparable to, or even less effective than plasma alone. These findings further underscore the dominant role of humoral factors in directly controlling *E. coli* proliferation in this system, with cellular functions contributing minimally to the initial innate immune response against bacterial infection (at least for complement-sensitive strains). Additionally, no significant differences in plasma bactericidal activity were observed across the age range studied.

## 5. Conclusions

Components of innate immunity appear to remain stable within our sample of junior and senior groups of people and monkeys, with no signs of increased genetic instability in our experimental conditions. SA-β-Gal activity was consistently high in all myeloid populations and did not vary within the age ranges studied. This suggests that the widely documented “myeloid bias” during aging is accompanied by preserved phagocytic capacity, unchanged basal production of proinflammatory cytokines, and stable activation of cytokines such as TNF and IL-6 in response to infection. Moreover, the sustained bactericidal function of plasma humoral factors ensures effective bacterial control even in older individuals, comparable to that observed in younger donors. Therefore, the observed quantitative increase in myeloid cells likely serves as a compensatory mechanism, with increased cell numbers potentially offsetting the decline in adaptive immunity and maintaining primary defense against bacterial infections.

The study also confirmed evolutionarily conserved features of innate immunity in humans and cynomolgus macaques, including phenotypic similarities between their principal cell populations and functions. Unexpectedly, prolonged retention of bacteria on the surface of *E. coli* and extended phagocytic activity of classical monocytes were observed in both species. While phenotypic differences in CD14 and CD16 expression between macaque and human granulocytes did not affect phagocytic activity, these phenotypic features should be considered when designing preclinical strategies targeting myeloid cells. Importantly, the stability of phagocytic function across age groups underscores the resilience of myeloid cells in both species. Additionally, the dominant contribution of humoral factors to bacterial control remains consistent regardless of cellular activity, ensuring maintained innate defense even as adaptive immune responses decline with age.

## Figures and Tables

**Figure 1 biomedicines-14-00071-f001:**
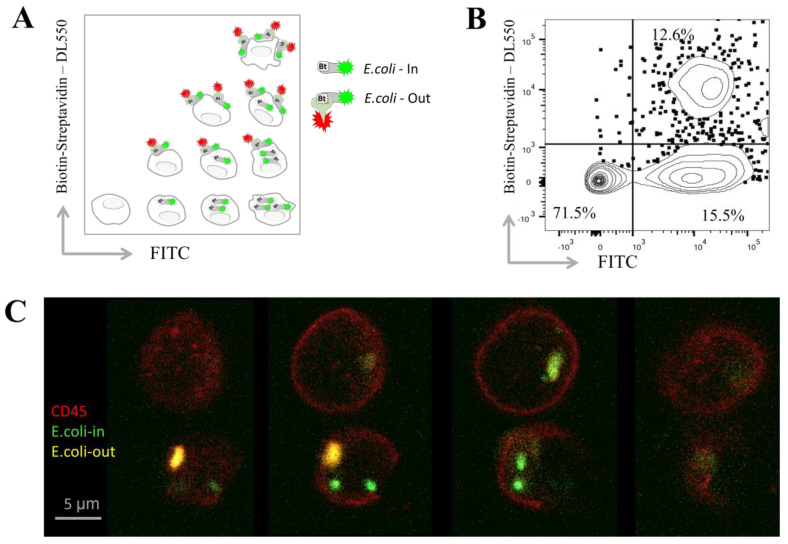
Double labeling of *E. coli* reveals the spatial localization of bacteria on the surface and inside myeloid cells after phagocytosis. (**A**) Schematic illustration of the distribution of phagocytes containing bacteria either inside or on the cell surface, based on multicolor flow cytometry. The diagram shows bacteria labeled with FITC and biotin/streptavidin-DL550. Bacteria inside phagocytes are labeled with FITC and biotin (Bt) (indicated as “in”), while bacteria on the surface are labeled with FITC and biotin+streptavidin–DL550 (indicated as “out”). (**B**) Representative experimental FACS-plot demonstrating the distribution of granulocytes based on fluorescence intensity, indicating bacteria located inside (FITC^+^) and outside (FITC^+^ and DL550^+^) the cells. (**C**) Confocal microscopy of phagocytes. Z-stacks of two cells are shown: the upper phagocyte contains bacteria inside (green/FITC), while the lower phagocyte shows one bacterium outside (green/FITC and streptavidin–DL550, merged as orange) and three bacteria inside (green/FITC). The surface of the phagocytes, stained with anti-CD45-APC-eFluor 780, is marked in red.

**Figure 2 biomedicines-14-00071-f002:**
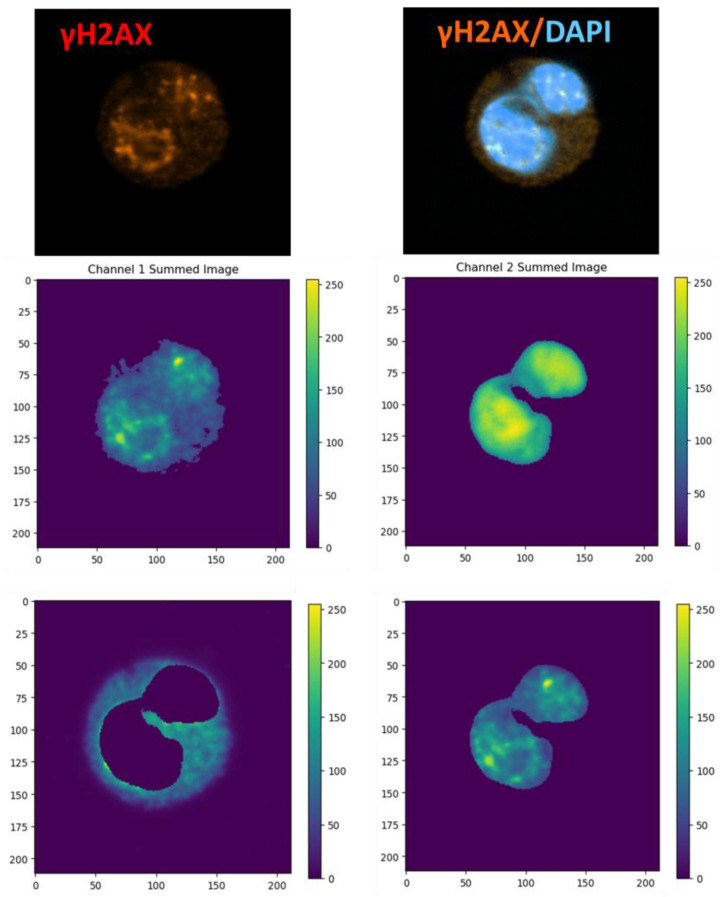
Example of mask for analysis of comparative staining for nucleus and cytoplasm.

**Figure 3 biomedicines-14-00071-f003:**
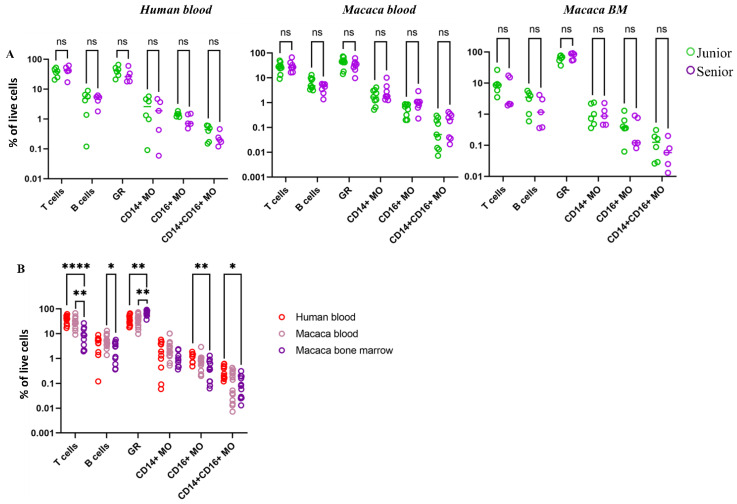
Flow cytometry of live singlet cells from human blood and in *M. fascicularis* blood and bone marrow. Panel (**A**) shows age-related changes in the frequencies of CD3^+^ T cells, CD20^+^/CD19^+^ B cells, granulocytes, and monocyte subsets (CD14^+^ single positive classical monocytes, ncMOs and inMOs) between junior and senior groups; Tissue types are indicated at the top of each graph. Panel (**B**) presents a comparison of these immune cell populations in blood and bone marrow samples from humans and *M. fascicularis*, regardless of age; *p* < 0.05 (*), *p* < 0.01 (**), *p* < 0.0001 (****), and ns (not significant).

**Figure 4 biomedicines-14-00071-f004:**
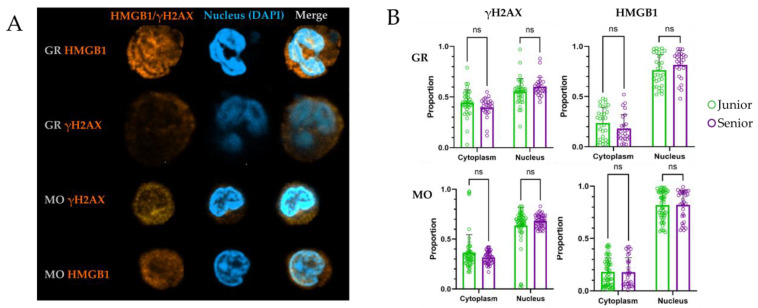
Confocal microscopy of HMGB1 and γH2AX localization. (**A**) Representative images of monocytes (MO) and granulocytes (GR). HMGB1 or γH2AX staining are highlighted in orange, the DAPI in blue, the overlay of fluorescent both markers is shown in white. (**B**) quantification of staining nucleus and cytoplasm between two age groups, n = 480 (30 cells of each type and each marker from each donor). Statistical difference analysis by Mann–Whitney U-test (ns—not significant).

**Figure 5 biomedicines-14-00071-f005:**
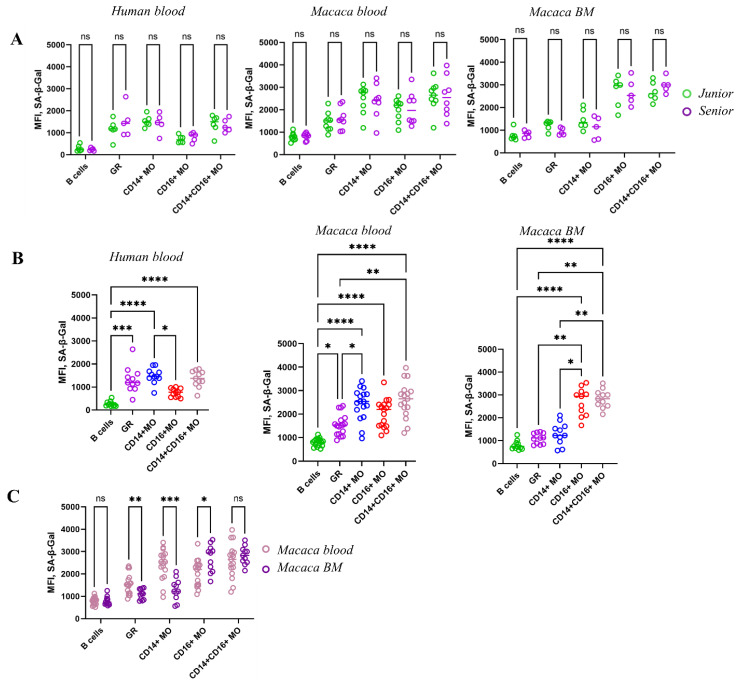
SA-β-Gal (SPiDER) mean florescence intensity (MFI) in whole blood and bone marrow cell populations of humans and *M. fascicularis* in two age groups,—junior and senior (**A**) and SA-β-Gal MFI comparison in different tissues in two species (**B**). Comparison of SA-β-Gal MFI between blood and bone marrow cell populations (**C**). The significance of differences was determined by the Mann–Whitney U-test; *p* < 0.05 (*), *p* < 0.01 (**), *p* < 0.001 (***), *p* < 0.0001 (****), and ns (not significant).

**Figure 6 biomedicines-14-00071-f006:**
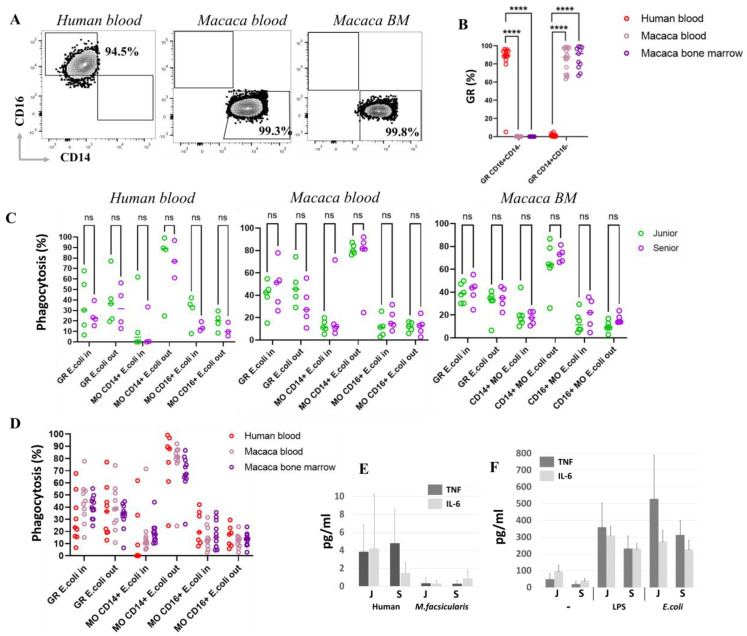
Phagocytosis of granulocytes (GR) and monocytes (MO) in CD14^+^ and CD16^+^ populations in human whole blood and cynomolgus monkey whole blood and bone marrow. Human and cynomolgus monkey granulocytes had different phenotypes (**A**), which shows a significant statistical difference between the species (**B**). The percentage of phagocytosed GR and MO populations that ingested *E. coli* internally (in) or captured them on the surface (out) is shown for two age groups (junior and senior) and for the two species. Statistical differences were calculated using Mann–Whitney multiple tests (**C**). Comparative analysis of phagocytosis in myeloid subpopulations for the two species is shown for the whole sample (**D**). Concentration of TNF and IL-6 in humans and *M.fascicularis* of two age groups, Junior (J) and Senior (S), in blood serum (**E**) and in the adhesive fraction of whole blood after stimulation with LPS or *E. coli* (**F**). N = 12. Distribution was not normal according to the Shapiro–Wilk test; statistical differences were assessed using the Kruskal–Wallis test. The tissue studied was indicated above the graphs and in the legends; *p* < 0.0001 (****), and ns (not significant).

**Figure 7 biomedicines-14-00071-f007:**
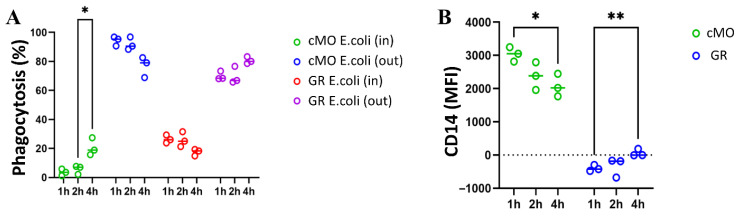
Time course of phagocytosis and CD14 surface expression in myeloid cells. (**A**) Percentage of phagocytic cells within human classical monocytes (cMOs) and granulocytes (GR) containing bacteria only internally (in) and with bacteria both inside and on the surface (out). (**B**) Mean fluorescence intensity (MFI) of the CD14 surface marker on phagocytes, measured after 1, 2, and 4 h of incubation with bacteria. Statistical differences were determined using Two-way ANOVA with multiple comparisons; *p* < 0.05 (*), *p* < 0.01 (**), and ns (not significant).

**Figure 8 biomedicines-14-00071-f008:**
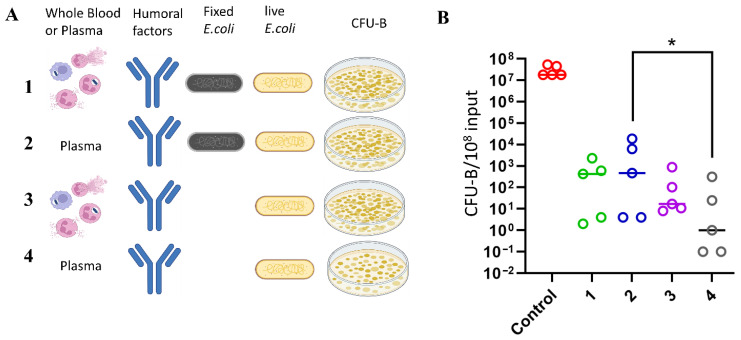
Comparison of antibacterial properties of whole blood and plasma. Experimental setup (for **A** and **B**): samples 1 (whole blood) and 2 (blood plasma) were pre-incubated for 4 h with fixed bacteria and then for 4 h with live bacteria. Samples 3 (whole blood) and 4 (blood plasma) were incubated for 4 h directly with live bacteria without the addition of killed *E. coli*. (**A**) Visual representation of the experimental setup. (**B**) Comparison of blood and plasma incubation options with the control (bacteria without incubation with blood components). All data are normalized to 10^8^ live *E. coli* input. CFU-B counts are presented on a logarithmic scale. Statistical analysis was performed using Friedman test (multiple comparisons); *p* < 0.05 (*).

## Data Availability

The original contributions presented in this study are included in the article and Appendix A. Further inquiries can be directed to the corresponding author.

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
