# Peer review of "Stability of Myeloid Cell Phenotype and Function Across a Broad Age Range in Humans and Cynomolgus Monkeys, and a Dominant Contribution of Humoral Factors in the Control of Bacterial Infection"

_biomedicines, 2025, doi:10.3390/biomedicines14010071_

Round 1

Reviewer 1 Report

Comments and Suggestions for Authors

This manuscript investigates the stability of myeloid cell phenotype and function across a broad age range in humans and cynomolgus macaques.  The topic of this article is meaningful, however, the discussion is thorough but could be more concise. Some background information could be shortened to allow more focused discussion of the novel findings. In addition, while complement and antibodies are mentioned, a deeper discussion of which specific factors might be responsible for the potent bactericidal effect observed and whether their activity is also age-stable would be valuable.

Author Response

Comments 1: This manuscript investigates the stability of myeloid cell phenotype and function across a broad age range in humans and cynomolgus macaques.  The topic of this article is meaningful, however, the discussion is thorough but could be more concise. Some background information could be shortened to allow more focused discussion of the novel findings. In addition, while complement and antibodies are mentioned, a deeper discussion of which specific factors might be responsible for the potent bactericidal effect observed and whether their activity is also age-stable would be valuable.

Response 1: Thank you for your positive feedback on our work. We have reduced some background information, including details about our methods in the Discussion section. At the same time, we have added information that helps further explain the meaning of our results (in the "Discussion" chapter).

Furthermore, we have added more information about humoral factors. For control samples, we inactivated serum at 56°C for 30 minutes (complement fixation test). As shown previously, this temperature inactivates the function of complement proteins but not antibodies. In this case, we did not observe any differences compared to bacteria incubated in LB medium optimized for bacterial growth. In conclusion, this strain is complement-sensitive, and we hypothesize that complement proteins have the greatest impact, at least on the viability of the E. coli DH5α strain.

Reviewer 2 Report

Comments and Suggestions for Authors

As a hematologist, I only have expertise to review Result 3.1 and 3.2 and related materials. The overall manuscript seems like a collection of loosely-related experiments without organic integration. 

Result 3.1: 

  • The wording in the title (and the overall title) is misleading. This study only provides data for proportions of subpopulations. It does not actually study the "stability of immunophenotype", which usually means the expression of markers on a population, such as HLA-DR, CD11b, and CD56 on monocytes, activation markers (such as CD38), large granular forms (CD57 and CD16 expression) or pan-T cell markers (CD2, CD3, CD5, and CD7) on T cells, as well as CD13, CD16, and CD11b on granulocytes.
  • Regarding the subpopulations, the data probably can also provide NK cell counts (lymphocyte gate, CD3 negative, CD16 or CD56 positive). 
  • The changes of proportions of classical (CD16-) and nonclassical (CD16+) monocytes are commonly seen in different human diseases. Here are some related literature (PMIDs: 29982327, 26358827) 

Result 3.2

Granulocytes and monocytes are usually short living cells (from marrow hematopoiesis to tissue sequestration or splenic removal), so no much active DNA damage is expected to observed. However, the aging blood cells may carry mutated DNA sequence inherited from the hematopoietic stem cells. (Please look up clonal hematopoiesis of indetermined potential [CHIP]). 

Author Response

Comments 1: As a hematologist, I only have expertise to review Result 3.1 and 3.2 and related materials. The overall manuscript seems like a collection of loosely-related experiments without organic integration. 

Result 3.1: 

  • The wording in the title (and the overall title) is misleading. This study only provides data for proportions of subpopulations. It does not actually study the "stability of immunophenotype", which usually means the expression of markers on a population, such as HLA-DR, CD11b, and CD56 on monocytes, activation markers (such as CD38), large granular forms (CD57 and CD16 expression) or pan-T cell markers (CD2, CD3, CD5, and CD7) on T cells, as well as CD13, CD16, and CD11b on granulocytes.
  • Regarding the subpopulations, the data probably can also provide NK cell counts (lymphocyte gate, CD3 negative, CD16 or CD56 positive). 
  • The changes of proportions of classical (CD16-) and nonclassical (CD16+) monocytes are commonly seen in different human diseases. Here are some related literature (PMIDs: 29982327, 26358827) 

Response 1: Thank you for your feedback. To avoid misleading we have changed “immunophenotype” to “phenotype”. We claim about stability only for myeloid cells, for this reason we do not consider T cells and NK cells in detail. We agree about using additional markers. Our results showed that CD38 is expressed transiently in human samples and may serve as a dynamic activation marker. Its expression level varies depending on donor conditions (e.g., infection). We did not include these data because donor-to-donor variations were greater than the age-related shifts. For these reasons, we consider this marker to be outside the main focus of the manuscript and not indicative of age-dependent alterations.

Since the main focus of our manuscript is a comparison of humans and monkeys, we selected antibodies that work in both species. We tested different antibody clones several times, and some markers had to be excluded from the study because they either worked only in humans or did not perform optimally in macaques. Additionally, in cynomolgus macaques, CD56 is expressed differently than in humans and does not mark differentiated NK cells. Despite the genetic similarity between macaques and humans, the phenotype of NK cells in macaques appears to be more complex than in humans, and in terms of aging, it has been poorly studied (PMID: 10451505; 23423644). Therefore, assessing additional aging component of the innate immune system (lymphoid-derived NK cells) would require a separate, expanded investigation.

In our future studies, we will expand the panel of antibodies suitable for both human and non-human primate samples in accordance with your comments.

Thank you for the references, we have added them to the “Discussion” section.

Comments 2: Result 3.2

Granulocytes and monocytes are usually short living cells (from marrow hematopoiesis to tissue sequestration or splenic removal), so no much active DNA damage is expected to observed. However, the aging blood cells may carry mutated DNA sequence inherited from the hematopoietic stem cells. (Please look up clonal hematopoiesis of indetermined potential [CHIP]). 

Response 2: Yes, monocytes and especially granulocytes are short-lived cells. Having studied their genomic integrity, we hypothesized that age-related genetic instability begins in early progenitors. We have now included this in the discussion to clarify our goals and objectives. We included this information in the Discussion section: “…CHIP is characterized by specific mutations in hematopoietic stem cells compartment and hence their expansion. Aging is a risk factor for CHIP development due to the accumulation of somatic mutations in hematopoietic stem cells. By analyzing myeloid cells from donors of junior and senior groups, we expected to observe differences that would be due to differences at the level of early precursors of these cells”.

Reviewer 3 Report

Comments and Suggestions for Authors

The manuscript presented by Elena V. Lysakova et. al. is a well-constructed and technically well-conducted study examining age-related changes in human and cynomolgus macaque myeloid populations using whole blood phagocytosis assays, senescence markers (γH2AX, HMGB1, SA-β-Gal), cytokine production, and plasma bactericidal tests.

The comparative primate approach, the inclusion of biologically old macaques, and the innovative dual-label bacterial assay represent notable strengths and offer a degree of novelty.

The experimental design is comprehensive and employs physiologically relevant conditions. The use of their own double labeling of E. coli to identify the spatial localization of bacteria on the surface and inside myeloid cells after phagocytosis is useful and provides relevant information about phagocytosis dynamics.

However, the human cohort is small and restricted to healthy, working-age individuals, limiting the generalizability of the broader conclusions regarding the absence of innate immune aging.

The use of a single complement-sensitive bacterial strain also constrains the interpretation of humoral dominance in antimicrobial defense. While the data convincingly show no detectable decline in the tested parameters within this cohort, conclusions occasionally overextend beyond what the dataset can firmly support.

Overall, the study is methodologically sound and technically rigorous, contributing valuable comparative insights into myeloid stability across adulthood. However, it would benefit from a more cautious interpretation of the findings and a more explicit acknowledgment of cohort and model limitations.

Considering that there exists evidence that most of the age-related biomarkers in immune cells showed nonlinear age trajectories—changing abruptly at specific ages rather than following a gradual, linear pattern over time¹, the conclusions presented in the article should be interpreted with caution.

So, I recommend putting attention on the following points:

1-Include a more complete table with demographic info:
Donor lifestyle
BMI
Metabolic status
Glucose, lipid, and cholesterol levels
Smoke and alcohol habits.

2-Be careful with the overgeneralization of “absence of immune aging” in myeloid cells
The Discussion frequently implies that myeloid cells do not undergo aging, which is indicated in several sentences:
“Myeloid cells and their genetic integrity remain remarkably stable across a broad age range” (p. 21).
“no signs of increased genetic instability” (p. 24).
“SA-β-Gal activity…did not vary with age” (p. 24).

This conclusion may be premature for several reasons:

The study includes healthy, employed adults only (22–73 years). As you note, this likely excludes frail, comorbid, or very elderly individuals (>80), where immunosenescence is most pronounced, or perhaps variation can occur in intermediate ages, such as 45 or 50.

The cohort size is small (n = 18 humans), which limits the power to detect moderate changes.
The age span is not linear: only two age groups are compared, and the “senior” human group peaks at ~73.

Only selected markers were assessed (γH2AX, HMGB1, SA-β-Gal), whereas other hallmarks of innate immune aging—such as ROS, telomere shortening, metabolic rewiring, epigenetic drift, and mitochondrial dysfunction—were not examined.

Finally, an inherent bias always exists when studying senescence, due to the possibility that people with most age-related alterations die before 60, for example.

My suggestion is to reword to indicate “within this cohort and under these experimental conditions, we did not detect evidence of innate immune decline.” Avoid implying a complete absence of innate immunosenescence. This observation includes both “Discussion” and “Conclusions” sections.

3- Improve the Discussion of species differences, especially indicating whether classical monocyte bacterial retention is identical in kinetics or magnitude between species, considering the differences in CD14/CD16 expression

4- I recommend conducting a deeper exploration of the classical monocyte bacterial-retention phenomenon, as this represents one of the most interesting and novel findings of the study. The discussion section could be strengthened by introducing hypotheses framed around guiding questions, such as:

Could it reflect delayed phagosome maturation, as seen in some monocyte subsets?
Do surface-bound bacteria remain viable, or is it possible that they are being degraded intracellularly?

5- Could be a good point to mention sample limitations and variability in the discussion section, for example:

The sample size
n=18 human donors
n=23 macaques
Number of cells analyzed for imaging-based markers
Sex differences (not discussed)
Variability in primate housing, diet, and microbial exposure
Single geographic location and limited genetic diversity

Add a paragraph in the Discussion addressing these limitations explicitly.

You may consider referring to a recent multi-omics aging study from Stanford University. It identified distinct molecular groups and showed that nearly 6,000 biomolecules exhibit nonlinear aging patterns. This could complement your Discussion by highlighting how aging processes often occur in waves rather than a gradual progression. Of course, this is only a suggestion if you feel it aligns with your narrative.

1Shen, X., Wang, C., Zhou, X. et al. Nonlinear dynamics of multi-omics profiles during human aging. Nat Aging 4, 1619–1634 (2024). https://doi.org/10.1038/s43587-024-00692-2

Finally, another minor suggestion and observations:

Page 1: Citation indicates Int. J. Mol. Sci. 2025, but the journal is Biomedicine.

Figure 2 B: The samples in this comparison belong to the entire cohort of participants? Or only the junior or senior groups?

Figures: Can the quality of the text in the images be improved? In some cases, we see text and numbers a little blurred.

Unify the use of classical monocytes, “cMO,” “CD14+CD16lo monocytes.” Choose one for all sections.

Author Response

Comments 1: The manuscript presented by Elena V. Lysakova et. al. is a well-constructed and technically well-conducted study examining age-related changes in human and cynomolgus macaque myeloid populations using whole blood phagocytosis assays, senescence markers (γH2AX, HMGB1, SA-β-Gal), cytokine production, and plasma bactericidal tests.

The comparative primate approach, the inclusion of biologically old macaques, and the innovative dual-label bacterial assay represent notable strengths and offer a degree of novelty.

The experimental design is comprehensive and employs physiologically relevant conditions. The use of their own double labeling of E. coli to identify the spatial localization of bacteria on the surface and inside myeloid cells after phagocytosis is useful and provides relevant information about phagocytosis dynamics.

Response 1: Thank you for your positive feedback about our work.

Comments 2: However, the human cohort is small and restricted to healthy, working-age individuals, limiting the generalizability of the broader conclusions regarding the absence of innate immune aging.

Response 2: Thank you, we softened the wording and mentioned that in a new section 2.13 "Limitations of the study".

Comments 3: The use of a single complement-sensitive bacterial strain also constrains the interpretation of humoral dominance in antimicrobial defense. While the data convincingly show no detectable decline in the tested parameters within this cohort, conclusions occasionally overextend beyond what the dataset can firmly support.

Response 3: Our initial hypothesis suggested that cellular factors, including phagocytosis, NETosis, and degranulation, would contribute significantly to bacterial killing. It is widely accepted that these cellular functions have a predominant impact in the control of bacterial infections. Taking into account that granulocytes and monocytes together form the main blood cellular population, we expected a visible reduction in bacterial load.

Apparently, granulocytes phagocytose already dead bacteria and thus do not contribute to the first line of defense (at least against bacteria E. coli, one of the most popular objects to study phagocytosis). It is possible that the complement reaction takes seconds, while cellular reactions take minutes. Therefore, the presence of cells in whole blood does not reduce the number of live bacteria. It is possible that blood cells are primarily busy engulfing bacteria destroyed by complement using complement receptors.

The protocols developed in this study may be applied to various bacterial strains, allowing for a more comprehensive assessment of the roles of cellular and humoral factors in antimicrobial defense. This point has been addressed in the newly added section on study limitations (section 2.13).

Comments 4: Overall, the study is methodologically sound and technically rigorous, contributing valuable comparative insights into myeloid stability across adulthood. However, it would benefit from a more cautious interpretation of the findings and a more explicit acknowledgment of cohort and model limitations.

Considering that there exists evidence that most of the age-related biomarkers in immune cells showed nonlinear age trajectories—changing abruptly at specific ages rather than following a gradual, linear pattern over time¹, the conclusions presented in the article should be interpreted with caution.

Response 4: Thank you for your positive feedback and helpful comments, which strengthen our study. We have softened the wording in the Introduction, Discussion, and Conclusion (we will discuss this in detail in the comments below). The nonlinear aging pattern is discussed in a new section 2.13, dedicated to the limitations of the study.

So, I recommend putting attention on the following points:

Comments 5: 1-Include a more complete table with demographic info:
Donor lifestyle
BMI
Metabolic status
Glucose, lipid, and cholesterol levels
Smoke and alcohol habits.

Response 5: Thank you, we have added this data to Supplementary Table S2. There is a reference in the text (section 2.2, the first paragraph): «The list of the donors and the additional information about donors’ health are attached in Supplementary data (see Supplementary Tables, S1 and S2).»

Comments 6: 2-Be careful with the overgeneralization of “absence of immune aging” in myeloid cells
The Discussion frequently implies that myeloid cells do not undergo aging, which is indicated in several sentences:
“Myeloid cells and their genetic integrity remain remarkably stable across a broad age range” (p. 21).
“no signs of increased genetic instability” (p. 24).
“SA-β-Gal activity…did not vary with age” (p. 24).

This conclusion may be premature for several reasons:

The study includes healthy, employed adults only (22–73 years). As you note, this likely excludes frail, comorbid, or very elderly individuals (>80), where immunosenescence is most pronounced, or perhaps variation can occur in intermediate ages, such as 45 or 50.

Response 6: Thank you, we have softened the wording and included the information about different age groups in the section 2.13 "Limitations of the study"

The cohort size is small (n = 18 humans), which limits the power to detect moderate changes.

Increasing the cohort size would only lead to an increase in the number of individuals outliers. Our goal was to identify universal markers of healthy immune aging. Most studies are based on clinical samples, which in turn creates a bias toward individuals with potential pathologies.

Furthermore, our findings are supported by studies in two primate species. We were truly surprised by the absence of significant attenuations in innate immunity even in a group of monkeys at the end of their lifespan (20-30 years old, equivalent to 66-100 years in human biological age). We incorporated these limitations into our study (section 2.13).

The age span is not linear: only two age groups are compared, and the “senior” human group peaks at ~73.

We agree, however, our goal was to identify early markers of immune system aging in order to find ways to prevent premature aging and slow the rate at which people enter the zone of exponential growth of pathological processes (we have added it to the discussion).

Only selected markers were assessed (γH2AX, HMGB1, SA-β-Gal), whereas other hallmarks of innate immune aging—such as ROS, telomere shortening, metabolic rewiring, epigenetic drift, and mitochondrial dysfunction—were not examined.

This is true. However, we also consider the number of functional declines (e.g., phagocytosis decline) and the control of processes such as telomere shortening, epigenetic ROS increase, and mitochondrial dysfunction, which will quickly be demonstrated in processes such as DNA double-strand breaks (γH2AX), the well-known indicator of cell stress, HMGB1, and SA-β-Gal, a known marker of cellular metabolic disorders and the accumulation of abnormal lysosomes. Only upon finding such hallmarks of cellular aging did we plan to move on to analyzing the mechanisms of these alterations. However, to our surprise, we did not find such alterations, so we did not investigate further dipper molecular mechanisms. We have added these comments to 2.13 section dedicated to the limitations of the study.

Finally, an inherent bias always exists when studying senescence, due to the possibility that people with most age-related alterations die before 60, for example.

Yes, we agree that this is a limitation of our research. Perhaps if we significantly increased the sample size, so that rare cases of premature aging could be identified and the subsequent fate of these individuals could be tracked, we could identify such individuals and detect signs of early health problems and accelerated aging of myeloid cells.

However, these outliers likely have various pathologies that cannot be grouped together (accelerated aging) and would require a different approach, such as utilizing specialized clinical samples. Our focus was on the course of healthy aging, which affects over 95% of the population, and, rather, their successful "immune strategies". For this reason, we considered a small sample size to be sufficient. We believe this sample is representative of working individuals and healthy aged monkeys.

My suggestion is to reword to indicate “within this cohort and under these experimental conditions, we did not detect evidence of innate immune decline.” Avoid implying a complete absence of innate immunosenescence. This observation includes both “Discussion” and “Conclusions” sections.

Thank you, we have softened the wording, emphasizing that our findings are based on our experimental conditions and limited sample size. We outlined the proposed restrictions in the section named “Limitations of the study”.

Comments 7: 3- Improve the Discussion of species differences, especially indicating whether classical monocyte bacterial retention is identical in kinetics or magnitude between species, considering the differences in CD14/CD16 expression

Response 7: We found no differences between species, analyzing populations of cMO in % of cells with engulfed/retained bacteria during 1 h (figure 6 D). Considering an interesting phenomenon of retaining bacteria by cMOs, we checked kinetics of this process, using human blood samples. Unfortunately, monkeys’ material is limited, and we do not provide this data in the study.

Comments 8: 4- I recommend conducting a deeper exploration of the classical monocyte bacterial-retention phenomenon, as this represents one of the most interesting and novel findings of the study. The discussion section could be strengthened by introducing hypotheses framed around guiding questions, such as:

Could it reflect delayed phagosome maturation, as seen in some monocyte subsets?

Response 8: Thank you very much, we were surprised when used our double-labeled bacteria. According to the literature, cMOs have a reduced capacity for phagocytosis but are capable of performing multiple phagocytic cycles in a row. Studying this phenomenon requires additional effort, including additional grant support. In this study, this was beyond the scope of our research because it was not associated with age-related changes.

Do surface-bound bacteria remain viable, or is it possible that they are being degraded intracellularly?

Thank you, this is an interesting question that we checked. We suppose that under the physiological conditions bacterial cells are immediately opsonized with complement proteins and antibodies. Opsonization prevents its spreading. Then, complement receptors and Fc-receptors of immune cells bond opsonized pathogens. We used E. coli and draw a conclusion about a dominant contribution of humoral factors in the immediate control of bacterial infection (section 3.5). However, after incubation with bacteria, we tried to wash unbound (not associated with the blood cells) bacteria out and analyze how many bacteria stay alive on cells. Despite this, we were not sure that we were able to wash cells out bacterial conglomerates which co-centrifuged with the blood cells (based on our control data). Nevertheless, the lack of statistical significance between serum and whole blood rather suggests that complement-killed bacteria adhere to cells and are phagocytosed while already dead.

Comments 9: 5- Could be a good point to mention sample limitations and variability in the discussion section, for example:

The sample size
n=18 human donors
n=23 macaques

Response 9: Thank you, we have added this information to the Study Limitations section (we also found and corrected an inconsistency: the study involved 25 human donors).

Number of cells analyzed for imaging-based markers

Thank you for this important note. We have added information about number of cells analyzed to the figure 4.

Sex differences (not discussed)

Thank you, we have added this information to a section named “Limitations of the study”

Variability in primate housing, diet, and microbial exposure

Thank you, we have added this information to section 2.2 named “Blood sampling”.

Single geographic location and limited genetic diversity.

All monkeys were originally imported from Southeast Asia and bred for a long time at a breeding facility in Sochi. In the revised version of the manuscript, this information is included in the "Materials and Methods" section.

Add a paragraph in the Discussion addressing these limitations explicitly.

In the chapter “Materials and Methods” we have added a new section – "Limitations of the study".

Comments 10: You may consider referring to a recent multi-omics aging study from Stanford University. It identified distinct molecular groups and showed that nearly 6,000 biomolecules exhibit nonlinear aging patterns. This could complement your Discussion by highlighting how aging processes often occur in waves rather than a gradual progression. Of course, this is only a suggestion if you feel it aligns with your narrative.

1Shen, X., Wang, C., Zhou, X. et al. Nonlinear dynamics of multi-omics profiles during human aging. Nat Aging 4, 1619–1634 (2024). https://doi.org/10.1038/s43587-024-00692-2

Response 10: Thank you, we have included this article in the “Limitations of the study” section (the first paragraph).

Comments 11: Finally, another minor suggestion and observations:

Page 1: Citation indicates Int. J. Mol. Sci. 2025, but the journal is Biomedicine.

Response 11: Thank you, we have corrected that.

Figure 2 B: The samples in this comparison belong to the entire cohort of participants? Or only the junior or senior groups?

The samples there belong to the entire cohort of participants. We found no differences between junior and senior groups inside the species (Figure 3 A, 6 C) and then compared all of the samples to find interspecies differences (Figure 3 B, 6 D).

Figures: Can the quality of the text in the images be improved? In some cases, we see text and numbers a little blurred.

Thank you, unfortunately I have seen reviews that MDPI system sometimes reduces quality. Our pictures have resolution at least 300 DPI. We will take this into account during the final layout.

Unify the use of classical monocytes, “cMO,” “CD14+CD16lo monocytes.” Choose one for all sections.

Thank you, we have corrected that.

Thank you for your detailed comments. We were thrilled that you understood our article so deeply and asked the same questions we were asking ourselves. Thank you very much for your work. Your comments have significantly strenghtened the manuscript. I hope this article will be of interest to readers.

Reviewer 4 Report

Comments and Suggestions for Authors

This is a very interesting and extensive study of the stability of myeloid cell immunophenotype and function in different age groups in humans and monkeys and of the dominant contribution of humoral factors in the control of bacterial infection.

The authors have performed several experiments using flow cytometry and confocal microscopy along with several other methods to prove the stability of myeloid cells and the contribution of humoral immunity in the control of bacterial infection. The study is well designed and clearly presented.

There are some really minor points to be addressed.

line 39 humans

line 31 for a long time

lines 46 and 48 what do you mean by conserved mechanisms of innate immune cell aging? Do you want to check only for the conserved ones or for possible differences? You could rephrase it.

line 110 findings

line 115 does not fully...

line139 was also used

The two last paragraphs of the introduction are the conclusions of the study

In 2.1 some words are missing Check it carefully

line 321 Table 1 I cannot find it.

line 323 S2 and S3. S1 is not mentioned.

line 497 What do you mean in the presence of naive and resident cells?

Author Response

Comments 1: This is a very interesting and extensive study of the stability of myeloid cell immunophenotype and function in different age groups in humans and monkeys and of the dominant contribution of humoral factors in the control of bacterial infection.

The authors have performed several experiments using flow cytometry and confocal microscopy along with several other methods to prove the stability of myeloid cells and the contribution of humoral immunity in the control of bacterial infection. The study is well designed and clearly presented.

Response 1: Thank you very much for positive comment of our study.

Comments 2 + Response 2: There are some really minor points to be addressed.

line 39 humans

Thank you, we have corrected that.

line 31 for a long time

Thank you, we have corrected that.

lines 46 and 48 what do you mean by conserved mechanisms of innate immune cell aging? Do you want to check only for the conserved ones or for possible differences? You could rephrase it.

Thank you for your comment. We agree that this was a "lost in translation" issue. We have rephrased the sentence as: “Furthermore, the evolutionarily conserved mechanisms underlying innate immune cell aging across different primate species are still poorly understood.” We have also corrected this throughout the manuscript.

line 110 findings

Thank you, we have corrected that.

line 115 does not fully...

Thank you, we have corrected that.

line139 was also used

Thank you, we have corrected that.

The two last paragraphs of the introduction are the conclusions of the study

Thank you for your comments. We have revised and shortened the text to:
"
Cynomolgus monkeys are one of the most widely used models in preclinical research, making this study relevant for a broad range of biomedical scientists. A comparison of myeloid populations in blood samples from humans and cynomolgus monkeys, including a previously unstudied group of elderly macaques over 20 years of age, revealed both interspecies similarities and differences in cellular aging processes among primates."

With your permission, we would like to retain this fragment to justify our choice of monkey biological material.

In 2.1 some words are missing. Check it carefully

Thanks for your comment, we have edited section 2.1.

line 321 Table 1 I cannot find it.

Thank you, we have replaced the reference to Table 1 with Supplementary Tables S3 and S4. We have also corrected the references to additional tables and figures throughout the text.

line 323 S2 and S3. S1 is not mentioned.

We corrected that (section 2.2 first paragraph; section 2.6 first and second paragraph; section 3.5 first paragraph).

line 497 What do you mean in the presence of naive and resident cells?

Thank you for noticing this, we have edited the end of section 3.1.

Thank you very much for your review.

Round 2

Reviewer 2 Report

Comments and Suggestions for Authors

I only have expertise in the immunophenotype part of this manuscript, which has been adequately addressed. Please refer to other reviewer for an overall evaluation. 

Reviewer 3 Report

Comments and Suggestions for Authors

The revised manuscript is clear and carefully written, and the authors have satisfactorily addressed the previous comments.

The revisions have improved the balance and interpretation of the results, resulting in a well-supported and appropriately delimited narrative.

Overall, the study provides valuable insights into myeloid cell stability across aging and is likely to be of interest to the journal’s readership.